

# Development and deployment of a mid-cost $CO_2$ sensor monitoring network to support atmospheric inverse modeling for quantifying urban $CO_2$ emissions in Paris

Jinghui Lian[1,2], Olivier Laurent[2], Mali Chariot[2], Luc Lienhardt[3], Michel Ramonet[2], Hervé Utard[1], Thomas Lauvaux[3], François-Marie Bréon[2], Grégoire Broquet[2], Karina Cucchi[1], Laurent Millair[1] and Philippe Ciais[2]

[1] Origins.earth, SUEZ Group, Tour CB21, 16 Place de l'Iris, 92040 Paris La Défense Cedex, France

[2] Laboratoire des Sciences du Climat et de l'Environnement (LSCE), IPSL, CEA-CNRS-UVSQ, Université Paris-Saclay, 91191 Gif sur Yvette Cedex, France

[3] Groupe de Spectrométrie Moléculaire et Atmosphérique (GSMA), Université de Reims-Champagne Ardenne, UMR CNRS 7331, Reims, France

*Correspondence to:* Jinghui Lian (jinghui.lian@suez.com) and Olivier Laurent (olivier.laurent@lsce.ipsl.fr)

**Abstract**. To effectively monitor the highly heterogeneous urban $CO_2$ emissions using atmospheric observations, there is a need to deploy cost-effective $CO_2$ sensors at multiple locations within the city with sufficient accuracy to capture the concentration gradients in urban environments. Its measurements could be used as input of an atmospheric inversion system for the quantification of emissions at the sub-city scale or separate specific sectors. Such quantification would offer valuable insights into the efficacy of local initiatives and could also identify unknown emission hotspots that require attention. Here we present the development and evaluation of a mid-cost $CO_2$ instrument designed for continuous monitoring of atmospheric $CO_2$ concentrations with a target accuracy of 1 ppm on hourly mean measurement. We assess the sensor sensitivity in relation to environmental factors such as humidity, pressure, temperature and $CO_2$ signal, which leads to the development of an effective calibration algorithm. Since July 2020, eight mid-cost instruments have been installed within the city of Paris and its vicinity to provide continuous $CO_2$ measurements, complementing the seven high-precision Cavity Ring-Down Spectroscopy (CRDS) stations that have been in operation since 2016. A data processing system, called CO2calqual, has been implemented to automatically handle data quality control, calibration and storage, which enables the management of extensive real-time $CO_2$ measurements from the monitoring network. Colocation assessments with the high-precision instrument show that the accuracies of the eight mid-cost instruments are within the range of 1.0 to 2.4 ppm for hourly afternoon (12-17 UTC) measurements. The long-term stability issues require manual data checks and instrument maintenance. The analyses show that $CO_2$ measurements can provide evidence for underestimations of $CO_2$ emissions in the Paris region and a lack of several emission point sources in the emission inventory. Our study demonstrates promising prospects in integrating mid-cost measurements along with high precision data into the subsequent atmospheric inverse modeling to improve the accuracy of quantifying the fine-scale $CO_2$ emissions in the Paris metropolitan area.

## 1 Introduction

Accurately and effectively monitoring $CO_2$ emissions from cities can provide valuable information for tracking progress in $CO_2$ emission reductions measures to achieve net-zero emissions (Seto et al., 2021). However, it remains challenging due to the large spatial and temporal variations and to sectoral diversity of the emission sources across urban environments. Combining atmospheric measurements, a high-resolution $CO_2$ emission inventory, an atmospheric transport model, and an optimization framework, atmospheric inversions of $CO_2$ fluxes over urban areas offer a new solution to monitor and verify $CO_2$ emissions in a timely manner (Turnbull et al., 2019; Lian et al., 2023). Existing top-down studies generally provide estimates of monthly budgets of fossil fuel $CO_2$ emissions at the whole city scale (e.g., Turnbull et al., 2011; Staufer et al., 2016) or at the district level (e.g.,





Lauvaux et al., 2020) using an atmospheric in-situ $CO_2$ monitoring network equipped with three to twelve sensors. An inversion system able to resolve emissions across different parts of the city or different sectors would bring more insights on the effectiveness of localized mitigation measures (e.g., low traffic emission zones, renovation of buildings in a specific district) and possible emission hotspots that could be targeted for cost-effective emission reductions (Gurney et al., 2015). However, increasing the

dimension of the inverse problem (due to the larger number of flux unknowns to solve for) will require additional information to determine the full complexity of emissions error covariances at high spatial and temporal resolutions (Lauvaux et al., 2020; Nalini et al., 2022). With the deployment of high-density observation networks, atmospheric measurements can provide a sufficient constraint to quantify $CO_2$ emissions at high spatial resolution, but also for different sectors if a sufficient level of accuracy and precision has been reached in the measured concentrations and atmospheric transport modeling (Wu et al., 2016; Turner et al.,

2020; Kim et al., 2022).

To overcome these limitations, significant investments have been made to increase both the spatial coverage and the frequency of $CO_2$ measurements. Innovative approaches for measuring fine-scale $CO_2$ concentrations in urban areas have been proposed and evaluated. These novel strategies include data collection with various methods. For instance, Mallia et al. (2020) used mobile measurements on a light rail public transit platform to quantify $CO_2$ emissions in Salt Lake City. Lian et al. (2019) introduced the

GreenLITE$^{TM}$ laser imaging system, which was deployed to measure $CO_2$ concentrations along 30 horizontal chords covering an area of 25 km$^2$ in the central Paris over a 1-year period in 2016. Recent endeavors in the conceptual design and deployment of low- and mid-cost (~10k€) sensors paved the way for dense networks of atmospheric $CO_2$ sensors within a city (e.g., Shusterman et al., 2016; Martin et al., 2017; Arzoumanian et al., 2019; Müller et al., 2020; Delaria et al. 2021). This type of $CO_2$ observing network consists of lower-cost medium-precision sensors that could be deployed at many places for high spatial and temporal density

sampling. Existing recommendations suggest an accuracy target for these sensors of 1 ppm on hourly mean measurement, which is suitable for urban atmospheric inversions (Wu et al., 2016; Turner et al., 2016).

In 2020, a pilot research and development (R&D) project for greenhouse gas (GHG) monitoring was carried out in the Paris metropolitan area through a collaboration between Origins.earth (https://www.origins.earth/), the Laboratoire des Sciences du Climat et de l'Environnement (LSCE) and la Ville de Paris. The Paris metropolitan area, also known as the Île-de-France region,

includes the city of Paris and its surrounding seven departments (Hauts-de-Seine, Seine-Saint-Denis, Val-de-Marne, Seine-et-Marne, Yvelines, Essonne and Val-d'Oise). As part of this project, nine mid-cost sensors were constructed and installed starting July 2020 to provide continuous $CO_2$ measurements with a target accuracy of 1 ppm on an hourly basis. Note that a ninth sensor was installed at Citylights in September 2023 and thus was not included in this study. These sensors were deployed in addition to the in-situ network of 7 high-precision Cavity Ring-Down Spectroscopy (CRDS) stations already operational within the city of

Paris and its surrounding since 2016. This CRDS instrument achieves an accuracy level of better than 0.1 ppm for one-hour average $CO_2$ concentration (Xueref-Remy et al., 2018). The combined CRDS and mid-cost network has been designed in order to determine variability of urban $CO_2$ emissions at finer spatiotemporal resolutions than ones using the CRDS network stations only (Lian et al., 2023).

The first objective of this study is to present the new mid-cost $CO_2$ instrument design (hereafter referred to as High-Performance

Platform (HPP) instrument), the characterization of the sensors in relation to environmental parameters in the laboratory, the calibration and quality control strategy to achieve the target accuracy of 1 ppm on hourly mean measurement, and the evaluation of the sensor performances. The second objective is to assess the contribution of these 8 mid-cost medium-precision HPP $CO_2$ instruments, together with the high-precision CRDS measurements and the high-resolution WRF-Chem model, for a better understanding of the spatiotemporal variations in $CO_2$ emissions and concentrations in the Paris region. Additionally, we discuss

the potential implications of assimilating both the medium- and high-precision in-situ $CO_2$ data into the atmospheric inversion





system, with the ultimate goal of increasing the accuracy of quantifying $CO_2$ emissions for Paris. This paper is organized as follows. In Section 2, we present the setup of the mid-cost $CO_2$ instrument, the necessary laboratory sensitivity tests and the calibration procedure derived from these tests. Section 3 begins with an assessment of the accuracy of the mid-cost instrument through its comparison to the collocated high-precision CRDS instrument. Following that, an analysis of the temporal and spatial patterns in

observed and modeled $CO_2$ concentrations is conducted. Conclusions and discussions are given in Section 4.

## 2 Methods

### 2.1 Instrument integration

Building upon the mid-cost HPP $CO_2$ measuring instrument described by Arzoumanian et al. (2019), an upgraded version which is more suitable for operations in the field has been developed. This new instrument combines an integrated $CO_2$ sensor unit with

gas container elements, all enclosed in a waterproof stainless-steel box with dimensions of L120 × W50 × H25cm and a weight of 41.7kg. The photos and schematics of this instrument are shown in Figure 1 and Figure S1, respectively. Compared to Arzoumanian et al. (2019), several improvements have been made to facilitate the transportation and maintenance of the instrument without interruptions in power, along with its utilization in field campaigns.

The integrated $CO_2$ sensor unit is contained in a plastic box with an inlaid liquid-crystal display (LCD) touch screen. The box

measures L30 × W30 × H17.5cm in dimensions and has an approximate weight of 8kg with all components included. Figure 1b provides an illustration of the box's internal components. It is mainly based on a commercial non-dispersive near-infrared (NDIR) $CO_2$ sensor with the HPP 3.2 version from Senseair AB, Sweden. The sensor measures the $CO_2$ mole fraction within the optical cell through the infrared light absorption following the Beer-Lambert law (Barritault et al., 2013). The HPP sensor is also equipped with a pressure sensor (LPS331AP, ST Microelectronics, Switzerland) that enables real-time data corrections. Due to the design

of the optical cell with open air exhaust, its internal pressure is close to the atmospheric pressure, even during the measurement of the gas tank (described below). Simultaneously, a SHT75 environmental sensor (Sensirion, Switzerland) is placed upstream the HPP sensor inlet in order to measure continuously relative humidity and temperature of the air sample. Furthermore, the sensor box is equipped with: 1) a switching AC-DC power supply converter that transforms 230V AC to 12V DC; 2) a one-hour Uninterruptible Power Supply (UPS) type battery that keeps sensor on during various maintenance tasks; 3) a Raspberry Pi3 model

B V1.2 that collects the data from all sensors; 4) a solenoid valve (SMC, Japan, model VDW250-6G-1-M5) that allows the connection of the sensor input to be swapped between the ambient air intake and a gas tank. The control of the solenoid valve is automatically managed by the Raspberry Pi3 according to the programmable defined sequence; 5) a diaphragm micro-pump (Gardner Denver Thomas, USA, model 1410VD/1.5/E/BLDC/12V) with a speed rate regulated by a dedicated controller board (specific design) using a mass flow meter (MFM) (SMC, Japan, model PFMV530-1) in order to continuously flush the

measurement cell at a constant flow rate of 1 liter per minute (LPM); 6) a membrane filter dedicated to removing particles from the ambient air. The plumbing design of the airflow inside this box is shown in Figure S1b. More details regarding the HPP sensor are described in Arzoumanian et al. (2019).

Another feature of the integrated $CO_2$ sensor unit is the addition of a 5L gas tank of dry compressed natural air, pressurized at 200 bars and calibrated in $CO_2$ (Figure 1a). This reference gas, so called target tank/gas, is injected once a day, at a fixed time in the

middle of the day, in order to correct the measurements for short-term drifts and variability throughout the deployment period (see $CO_{2\,offset}$ in Eq.1). The tank is filled with dry ambient air and its $CO_2$ mole fraction, ideally close in concentration to the typical $CO_2$ mole fraction observed onsite during the afternoon, is assigned using a calibrated reference CRDS instrument (Picarro G2401) at LSCE before being installed at the sites. A linear interpolation between two successive injections of dry gas is applied. This



method typically results in a complete tank lifespan of approximately 5 months. A flushing pump (Figure 1a) could be optionally installed upstream the integrated $CO_2$ box in order to increase the flow rate and thus decrease the residence time in the sampling system when site configuration requires a long sampling line (EATON Synflex 1300). The box is equipped with a pair of fans to vent the equipment and avoid overheating especially in summertime, and two power supply strips. One strip serves the above

mentioned HPP $CO_2$ sensor box, while the other serves to the flushing pump.

## 2.2 Laboratory tests

The HPP sensor has a pre-calibrated factory configuration designed to measure $CO_2$ within the range of 0 to 1000 ppm. However, the raw $CO_2$ mole fraction directly reported by HPP is influenced by environmental factors, specifically water vapor ($H_2O$), pressure ($p$) and temperature ($T$). In order to achieve the target accuracy of 1 ppm on hourly basis, it is essential to carry out

sensitivity tests for every HPP sensor as demonstrated in previous studies (Arzoumanian et al., 2019; Liu et al., 2022). We conducted a series of laboratory tests (Table 1) for 8 integrated HPP $CO_2$ sensor boxes shown in Figure 1b. These tests are critical for establishing the specific correction coefficients for each sensor concerning its $CO_2$ sensitivity with respect to variations in $H_2O$ mole fraction, $p$, $T$ and $CO_2$ mole fraction. These correction coefficients will subsequently be used in the calibration equation Eq. (1) to calibrate the HPP $CO_2$ measurements, as detailed in Section 2.2.3. The magnitudes of the corrections for these four parameters,

aimed at reducing the bias of hourly $CO_2$ mole fractions, are presented in Section 3.1.

### 2.2.1 Water vapor sensitivity test

The water vapor sensitivity test was carried out at Atmosphere Thematic Centre (ATC) Metrology Lab (MLB) at LSCE. It consists of humidifying the dry natural air (containing $CO_2$ at ambient level, typically around 420 ppm) from a gas cylinder to various levels of water mole fractions ranging from 0% to 2.5%v with increments of 0.5%v. This is achieved by using a humidifying setup

that includes a liquid Mass Flow Controller (MFC) and a gas MFC, both supplying an evaporator chamber as shown in Figure S2a. The HPP sensor measures each step of the water vapor test for 10 minutes, and the entire sequence was repeated three times, resulting in a total duration of 3 hours. The $H_2O$ correction is used to provide the mole fractions in dry air from the raw measurements done in wet air conditions. We applied a quadratic polynomial fitting of the ratio between wet and dry $CO_2$ mole fractions in relation to water mole fractions (Figure S3a). This water vapor correction takes into account water vapor dilution effect

and any spectroscopic effect. The derived regression coefficients will serve as the correction factors in Eq. (1) to adjust the impacts of water vapor during the $CO_2$ data calibration process.

### 2.2.2 Pressure and temperature sensitivity test

Temperature and pressure sensitivity tests were performed to evaluate the response of each sensor to $CO_2$ under different $T$ and $p$ conditions. These experiments were carried out in a closed climatic chamber using the Plateforme d'Integration et de Tests (PIT)

at the Observatoire de Versailles Saint-Quentin-en-Yvelines (OVSQ) in Guyancourt, France. Figure S2b shows the schematic of this sensitivity setup. The methodology involves varying one of the two parameters while keeping the other constant and measuring the $CO_2$ mole fraction from indoor air within the climatic chamber with a sufficiently stable value of approximately 450 ppm. Parallel to the HPP instruments in the climatic chamber, a high-precision CRDS (Picarro G2401) sensor is placed outside the chamber with an inlet measuring the air inside the chamber to serve as a reference. To ensure a stable and homogenized air

condition in the climatic chamber, we set the temperature to 15°C and the pressure to 975hPa, and maintained it for a duration of 12 hours before the sensitivity test. For pressure sensitivity tests, the pressure was adjusted with six stages of (800hPa, 975hPa, 925hPa, 975hPa, 900hPa, 975hPa), while the temperature was held constant at 15°C. During temperature sensitivity tests, the



temperature was adjusted with six stages of (-10℃, 25℃, 40℃, 5℃, 30℃, 15℃), while maintaining a constant pressure of 975hPa. Each step of the pressure/temperature test lasted for 50 minutes, with the values undergoing a slow linear change between the two preset constant values for the first 30 minutes and remaining constant for the next 20 minutes. We repeated this entire sequence three times, resulting in a total duration of 15 hours. Initially, we corrected the $CO_2$ mole fraction readings obtained from the

sensors for potential water effects, despite the relatively low water concentration within the chamber. This correction was based on the coefficient derived from the prior water vapor sensitivity test. Subsequently, we applied a linear fit between the changes in pressure and $\Delta CO_2$ (differences in $CO_2$ mole fractions between CRDS and HPP measurements) (Figure S3b). As for the temperature, a quadratic polynomial fit was found to be more suitable than a linear regression, and thus was used (Figure S3c). The derived coefficients are used as correction factors in Eq. (1) to fix the impacts of variations in pressure and temperature on

$CO_2$ values reported by each HPP instrument.

### 2.2.3 Calibration procedure

The calibration procedure is necessary to ensure accurate $CO_2$ measurements by aligning the instrument readings with an official scale (e.g., WMO $CO_2$ scale). We tested the sensitivity of HPP to $CO_2$, by measuring dry air from two target cylinders with known $CO_2$ mole fractions of 400 and 5000 ppm. By using two mass flow controllers, we mixed the dry gas with a high $CO_2$ mole fraction

(5000 ppm) with the dry gas with a standard mole fraction (400 ppm), creating a sequence of seven mole fraction steps over the 400-600 ppm range (Figure S3d). To mitigate delays in HPP3 responses and ensure stability following thorough $CO_2$ flushing of each sensor cell, we sequentially sampled each mole fraction for a duration of 10 minutes, utilizing only the last 3 minutes of data. These measurements are performed in parallel with a calibrated high-precision CRDS instrument (Picarro G2401) that is used as a reference. The accuracy of this CRDS instrument calibrated with standards traceable to the WMO $CO_2$ X2019 calibration scales

is lower than 0.1 ppm (Hall et al., 2021). Figure S2c shows the schematic of this experiment setup. This $CO_2$ sensitivity test is periodically redone when the sensor is returned to LSCE for maintenance to ensure the accuracy of the coefficients.

Based on the aforementioned sensitivity tests, the calibration strategy consists of applying correction coefficients obtained by the influence of $T$, $P$ and $H_2O$ in the following calibration equation Eq. (1).

$$CO_{2_{cal}} = \left( \frac{CO_{2_{raw}}}{1 + IH_1 \times H_2O + IH_2 \times H_2O^2} - IT_1 \times (T - 26) - IT_2 \times (T^2 - 26^2) - IP_1 \times (P - 1015) \right) \times IC_1 + CO_{2_{offset}} \qquad (1)$$

In which, $CO_{2_{raw}}$ is the raw $CO_2$ mole fraction reported by the HPP sensor. $IH_1$ and $IH_2$ are the water vapor correction factors obtained from the water vapor sensitivity test. $H_2O$ is the water vapor mole fraction calculated from the Rankine's formula (Bérest and Louvet, 2020) which uses the pressure, temperature and relative humidity measured by the HPP and SHT75 sensors. $IT_1$ and $IT_2$ are the temperature correction factors derived from the temperature sensitivity test. $T$ is the temperature measured by the SHT75 sensor. $IP_1$ is the pressure correction factor obtained from the pressure sensitivity test. $P$ is the pressure measured by the

HPP sensor. Note that the corrections for $T$ and $p$ are made based on empirical equations by utilizing values of 26℃ and 1015 hPa, respectively. During the calibration period, $CO_{2_{cal}}$ in Eq. (1) represents the $CO_2$ mole fraction measured by the reference CRDS instrument calibrated on the WMO $CO_2$ X2009 scale (Hall et al., 2021). The $CO_2$ correction coefficient $IC_1$ is determined through a multipoint $CO_2$ regression using the seven mole fraction values assigned within the 400-600 ppm range during the $CO_2$ sensitivity test and initial lab calibration phase mentioned above. $CO_{2_{offset}}$ is the offset correction to rectify the instrument drift of the minute

$CO_2$ sampling data, which is determined from a linear interpolation between two successive daily target gas injections. The target gas is injected each day at a fixed time during midday for a duration of 3 minutes and only the last-minute data are used in the $CO_{2_{offset}}$ calculation.



### 2.2.4 Colocation evaluation

Once the correction and calibration coefficients are established (Table S2), Equation (1) could be applied to the raw HPP measurements to provide the corrected and calibrated $CO_2$ values. To evaluate the calibration quality and the performance of each integrated HPP instrument in real field conditions, a colocation experiment was carried out on the rooftop of the LSCE laboratory

building at an elevation of about 14 meters above street level. During this phase, $CO_{2_{cal}}$ in Eq. (1) represents the calibrated HPP $CO_2$ mole fraction. We then calculated the root mean square error (RMSE) of the $CO_2$ differences ($\Delta CO_2$) between the calibrated HPP mole fractions and the reference data collected simultaneously by the CRDS instrument during this colocation period lasting at least two weeks.

The initial assessment of the colocation performance took place before deploying the HPP instruments in the Paris monitoring

network. Meanwhile, in order to improve the quality control, this evaluation was also carried out for every replacement of the 5L target tank. When the pressure in the target tank drops below 20 bars (approximately every 4-5 months), the HPP instrument is returned to LSCE for tank replacement. Once installed at the LSCE rooftop, a 3-day colocation evaluation with a reference CRDS instrument is conducted with the tank which was currently used on site and then the new tank which will be used. The primary goal of the first 3-day (minimum) colocation is to check the HPP performance with the tank used on site and the need for calibration

coefficient updates (which require a longer colocation period indeed, and even additional tests in laboratory). On the other hand, the subsequent 3-day colocation is intended to confirm the instrument performance with a new tank before its reinstallation for onsite use.

### 2.3 Data processing chains

Data calibration has been centralized in order to ease future evolution of the calibration process and enable raw data storage

redundancy. A $CO_2$ data processing system named CO2calqual has been implemented, which utilizes cloud architecture for automated data quality control, calibration and storage for the HPP monitoring network (Figure 2). The raw data (e.g., $CO_2$, pressure, temperature and relative humidity) measured by each HPP station are automatically sent to the CO2calqual server's SFTP using 3G/4G LTE connections on a daily basis. Following this, the data are transferred via a Microsoft Azure synapse pipeline to a centralized data warehouse hosted on an Azure Blob storage, where all materials are carefully saved in a read-only archive. These

newly collected raw data are processed once a day by a server hosting the CO2calqual calibration algorithm implemented in the form of a Python library. This subroutine comprises three crucial procedures namely data calibration, quality control and time aggregation, which are presented in Sections 2.2, 2.3.1 and 2.3.2 respectively. Finally, the processed data are archived in the CO2calqual database and made available for subsequent usage.

### 2.3.1 Quality control

Data quality control (QC) is implemented in the CO2calqual system to account for potential issues arising from physical sensor malfunctions and local sources of contamination in urban environments. First, the automatic QC of the raw data may identify and flag out erroneous data because of incorrect or abnormal internal physical parameters of the HPP instrument. These incorrect physical parameters include, but are not limited to, factors such as the temperature or pressure of the instrument cell being outside its valid range. A list of internal flags for some important physical parameters could refer to Table S1. Additionally, regular manual

quality control on corrected/calibrated data is also performed based on the expertise of the station principal investigator and information documented in the instrument maintenance logbook. In order to maintain consistency with the CRDS data, we adopt the same data flagging labels used by the ICOS ATC system (Hazan et al., 2016). More specifically, letters U and N are used to



flag the valid and invalid data in the automated quality control process respectively, while letters O and K are used for the same purpose in the manual quality control process.

Second, a spike detection algorithm is implemented to identify potential local sources of contamination in the continuous time series of $CO_2$ data. The algorithm is based on the standard deviation method following El Yazidi et al. (2018) with minor parameter
adjustments by trial and error to accommodate increased variability of the $CO_2$ signal in urban environments. The equation Eq. (2) is given below:

$$C_i \geq C_{unf} + \alpha \times \sigma + \frac{\sqrt{n}}{10} \times \sigma \qquad (2)$$

In which, $C_i$ is the minute $CO_2$ data to be tested. It will be identified as a spike if its value is larger than the threshold specified on the right side of the equation. $C_{unf}$ is the last $CO_2$ data considered as non-spike. $\alpha$ is a parameter to control the selection threshold.
Same as El Yazidi et al. (2018), $\alpha$ was set to 1 for $CO_2$. $\sigma$ is the computed standard deviation on two middle quartiles over 1-week time windows. $n$ is the number of $CO_2$ data between $C_i$ and $C_{unf}$. The minute $CO_2$ data will then be categorized as either "spike" or "non-spike". Afterward, we also calculate and label the fraction of spikes within each hour based on these minute-level data.

### 2.3.2 Time aggregation

The HPP data undergo two stages of time aggregation within the CO2calqual system. The initial time aggregation involves
consolidating raw measurement data collected from HPP sensors, which are sampled approximately every second. These data are averaged at the temporal resolution of one minute, after which the calibration procedure is applied. Further temporal averaging at the hourly time scale is applied on the calibrated data. Note that the averaging uses only the valid data (those with either a N or K flag are excluded).

### 2.4 Instrument deployment

The HPP instruments have been gradually deployed on-site for a continuous field measurement since July 2020. Figure 3 shows the locations and photos of the eight HPP and seven CRDS $CO_2$ monitoring stations in the Paris region, together with their installation dates. The HPP stations are roughly distributed in the northwest-southeast direction, serving as a complement to the previous CRDS stations in the northeast-southwest direction (the prevailing wind direction over the Paris region). Additionally, they are located closer to the city center of Paris, facilitating improved monitoring of urban $CO_2$ signals. The selection of sampling
sites primarily adhered to the following criteria: 1) stipulating a minimum building height of > 15 meters, 2) ensuring the building surpasses its neighboring structures in height, 3) confirming the building relies mainly on electricity or has no energy source, 4) maintaining minimal daily occupancy to mitigate exhaust contaminations, 5) the building is located at a distance from high-emission sources, and 6) facilitating easy authorization for rooftop sensor installation. Finally, the instruments were deployed on carefully vetted high-rise buildings, positioned at different elevations ranging from 16 to 165 meters above ground level (Table 2).
The identification numbers of HPP (1 to 8) instruments in the laboratory and their corresponding installation site names are given in Table 3.

### 2.5 WRF-Chem model setup

The atmospheric transport model links $CO_2$ emissions to atmospheric concentrations. It represents the processes that lead to dilution and mixing of $CO_2$ emissions, thereby enabling the interpretation of $CO_2$ concentration measurements and how they relate to
emissions. In this study, $CO_2$ observations from the eight HPP and seven CRDS $CO_2$ stations are compared with outputs of the Weather Research and Forecasting model coupled with Chemistry (WRF-Chem) V3.9.1 transport model (Grell et al., 2005) at 1-km horizontal resolution. The setup of the WRF-Chem model is described in detail in Lian et al. (2019) and is outlined briefly here.





The fossil fuel $CO_2$ emissions were taken from a 1-km gridded hourly inventory produced by Origins.earth (Lian et al., 2022, 2023). The biogenic $CO_2$ fluxes were simulated by the Vegetation Photosynthesis and Respiration Model (VPRM) online-coupled with WRF-Chem (Mahadevan et al., 2008). The meteorological and $CO_2$ initial and lateral boundary conditions were retrieved from the global European Centre for Medium-Range Weather Forecasts (ECMWF) Reanalysis v5 (ERA5) dataset (Hersbach et al.,

2020) and the Copernicus Atmosphere Monitoring Service (CAMS) near-real-time $CO_2$ dataset, respectively. The simulations were performed for a duration of 29 months from July 2020 to December 2022, covering the entire period of the $CO_2$ measurements analyzed in this study.

## 3 Results

### 3.1 HPP instrument performance

**3.1.1 Sensitivity to environmental factors**

Table 3 summarizes the derived regression coefficients utilized in the $CO_2$ calibration equation (Eq. 1) for the correction due to environmental factors ($H_2O$, $p$, $T$ and $CO_2$ mole fraction) for each HPP sensor. These coefficients are determined through the laboratory sensitivity tests detailed in Section 2.2. As an illustrative example, Figure S3 shows the relationships between the raw 1-minute averaged $CO_2$ mole fraction reported by one of the HPP sensors (HPP3) and variations in $H_2O$, $p$, $T$ and $CO_2$ mole

fraction in the sensitivity tests, respectively. Similarly, the regression results for all the 8 HPP sensors are presented in Table 3. The $H_2O$ sensitivity test shows a sensor-specific response, where $IH_1$ values span from -3.92×10⁻³ to 0.75×10⁻³ ppm/%v and $IH_2$ values range from -2.18×10⁻³ to -0.48×10⁻³ ppm/%v. After the $H_2O$ correction, the $CO_2$ mole fractions reported by HPP sensors have residual deviations less than ±0.5 ppm relative to the assigned dry air mole fraction in the target cylinder. The $CO_2$ sensitivity to $T$ changes is also dependent on the sensors and ranges from -5.69 to 1.14 ppm/°C. After the $T$ correction, $CO_2$ mole fractions

reported by HPP sensors exhibit $R^2$ of 0.804~0.995 when compared to the reference $CO_2$ values measured by the reference CRDS instrument in the temperature sensitivity tests. Conversely, the variations in $CO_2$ mole fractions due to $p$ changes exhibit a similar magnitude across different sensors with a narrow range of 0.055 to 0.065 ppm/hPa. After the $p$ correction, $CO_2$ mole fractions from HPP sensors have $R^2$ of 0.998~0.999 against the reference instrument in the pressure sensitivity tests. The $CO_2$ correction coefficients for different sensors closely approach 1, ranging from 0.997 to 1.075.

Figure 4 shows the hourly time series of $CO_2$ mole fractions measured by each HPP sensor and the collocated reference measurements at LSCE laboratory rooftop during a colocation period of 3~11 days. It gives the magnitudes of the corrections for each component in Eq. (1). Note that corrections accumulate in sequence in Eq. (1), starting with $H_2O$, followed by $T$, $p$ and $IC_1$, and finally $CO_{2_{offset}}$. We computed the RMSE values of hourly $\Delta CO_2$ mole fractions between the calibrated HPP $CO_2$ data and the reference $CO_2$ values obtained from the CRDS instrument. This calculation was performed for both the afternoon period (12-

17 UTC) and the entire day. Results show that in the absence of any corrections nor calibration, the RMSEs of the all-day hourly $\Delta CO_2$ vary from 9.3 ppm (HPP4) and 58.8 ppm (HPP5). When applying the $H_2O$ correction, the RMSEs of $\Delta CO_2$ slightly change at a magnitude of -1.4~4.3 ppm. The daily variations of $CO_2$ mole fractions are noticeably improved after applying the $T$ and $p$ correction, e.g., on December 11ᵗʰ and 13ᵗʰ 2020 at HPP4, and on December 4ᵗʰ 2020 at HPP5. The $p$ correction substantially reduces the RMSEs of $\Delta CO_2$ to 1.6 ppm (HPP4) to 49.7 ppm (HPP7). The RMSEs after the $CO_2$ correction ($IC_1$) vary from 2.1

ppm (HPP1) to 21.8 ppm (HPP4). Finally, the daily injection of the target tank ($CO_{2_{offset}}$) significantly reduces the RMSEs to 0.9 ppm (HPP1) to 2.7 ppm (HPP6 and HPP7). Furthermore, among the eight HPPs, five show that the calibrated $CO_2$ mole fractions in the afternoon align more closely with the reference data than the all-day hourly data, exhibiting RMSEs ranging from 0.3 ppm





(HPP3) to 2.6 ppm (HPP6). Our results indicate that although the other corrections ($H_2O, T, p, IC_1$) provide improvements of the HPP sensor, the instrument needs a target gas to achieve its optimal performance.

### 3.1.2 Colocation performance

The performance of each HPP instrument is evaluated during the colocation period with a high-precision CRDS instrument (Picarro G2401) as described in Section 2.2.4. Figure 5a shows the differences in hourly afternoon (12-17 UTC) $CO_2$ mole fractions between the calibrated HPP data and the collocated CRDS measurements during all the intercomparison periods varying from 45 to 124 days. The median values of the hourly afternoon $\Delta CO_2$ mole fractions between HPP and CRDS instruments fall within the range of -1.1 to 1.7 ppm. Each of the eight HPP instruments demonstrates its individual accuracy. Five of them have RMSE values less than or equal to 1.5 ppm. HPP2 performs the best with an RMSE of 1.0 ppm, while HPP6 has the least favorable performance with

an RMSE of 2.4 ppm. When considering other times of the day (18-11 UTC), the differences between HPP and CRDS measurements during colocations have slightly larger RMSEs ranging from 1.3 to 3.9 ppm (Figure 5b). This is because $CO_2$ concentration in the target tank is close to the afternoon ambient levels and is much lower than most of the concentrations at nighttime. The target gas injection is scheduled for midday each day, allowing for more effective correction of data measured around that time in similar environmental conditions and also with smaller drifts in time between two consecutive daily injections.

Indeed, the $CO_{2_{offset}}$ correction from the target gas injection allows to correct the sensor intrinsic variability and drift, but also correct residuals from the $p$ and $T$ correction applied for the conditions encountered at the time of the gas injection which might be representative to the afternoon conditions. In consequence, the offset correction can compensate for the imperfection of the $p$ and $T$ correction for the few hours surrounding the injection time but might not compensate for different conditions ($p, T, CO_2$) such as the residual from the effect temperature diurnal cycle at nighttime. It should be noted that the offset correction is not able

to compensate for the imperfection of the water vapor correction as the gas from the tank is dry.

Figure S4a shows that the median concentrations of local simulated hourly afternoon (12-17 UTC) $CO_2$ signals, originating from both fossil fuel and biogenic sources in the Paris metropolitan area, exceed 2 ppm, with standard deviations ranging from 7.9 to 12.2 ppm. The standard deviations of model-observation misfits in hourly afternoon $CO_2$ mole fractions at each HPP station are greater than 7.4 ppm (Figure S4b). A prior sensitivity test also shows that the differences in simulated hourly afternoon $CO_2$ mole

fraction between two fossil fuel $CO_2$ emission inventories have standard deviations ranging from 3.2 to 8.2 ppm (Figure S4c and Lian et al., 2023). These results demonstrate that both the local $CO_2$ signals and the uncertainty in fossil fuel $CO_2$ emissions exhibit significantly greater magnitudes compared to the accuracy of HPP instruments (Figure 5). It indicates that the HPP instrument is able to provide valuable information for $CO_2$ monitoring following its on-site deployment, with the ultimate goal of revealing the distribution of $CO_2$ emissions in the Paris metropolitan area.

### 3.2 Model-data comparison

### 3.2.1 $CO_2$ concentrations

Figure S5 shows the data availability of the observed hourly $CO_2$ mole fractions at each station, together with the simulated $CO_2$ mixing ratios reproduced by the WRF-Chem model over the entire study period spanning from July 2020 to December 2022. In general, the observations and the modeled $CO_2$ mole fractions are in fairly good agreement, showing seasonal variations and their

correlation with atmospheric processes that influence the evolution of the planetary boundary layer (PBL). Notably, higher peak $CO_2$ mole fractions are often observed during the winter and nighttime, particularly at stations close to the city of Paris. It is worth noting that intermittent data gaps occurred at each station, lasting for several days to several weeks (Figure S5). The percentages



of valid hourly $CO_2$ observations after quality control to the total theoretical observational hours since site establishment range from 52% at DEF to 83% at OBS. The data gaps primarily stemmed from instrument failures, power outages, 3G/4G data transfer issues, and extended periods of instrument maintenance. This indicates that the lower-cost instruments do not exhibit the same level of stability as the CRDS instruments (> 91%) when used for long-term continuous outdoor measurements. Therefore, it

demonstrates the importance of automatically detecting data loss, promptly pinpointing its causes, and improving the efficiency of instrument maintenance when managing a large number of instruments within the urban $CO_2$ monitoring network.

Figure 6 shows the distributions and statistics of the observed and modeled hourly afternoon (12-17 UTC) $CO_2$ mole fractions, as well as the model-observation misfits at the 7 CRDS Picarro stations and 8 HPP stations (Figure 3) respectively over the period of July 2020 to December 2022. To ensure comparability in model-data comparisons, we applied data filtering to the simulated $CO_2$

mixing ratios by retaining data only when a corresponding valid observation was available at the same time. In Figure 6, both CRDS and HPP stations are displayed in a top-to-bottom sequence, corresponding to their increasing distance from the JUS station located in the center of Paris. Results show that the distributions of observed and simulated $CO_2$ mole fractions at different stations exhibit a rough similarity, and the ranking of average $CO_2$ mole fraction values at these stations is also generally consistent between the measurement and the model. In general, the HPP instruments exhibit similar magnitudes of model-observation misfits in $CO_2$

mole fraction when compared to CRDS instruments (Figure 6c and Figure S6). This to some extent further implies that HPP may not have large measurement errors. The $CO_2$ mole fractions observed and simulated at various HPP stations (with the median plus/minus standard deviation varying from 420.2±13.6 ppm to 426.2±16.9 ppm) tend to be higher than those at CRDS peri-urban sites (varying from 416.2±11.7 ppm to 422.9±14.7 ppm). This is because most of HPP stations are located in proximity to the city of Paris where anthropogenic $CO_2$ emissions are densely concentrated and higher than in the surrounding. Conversely, in the case

of CRDS stations, only JUS and CDS are urban stations, with the remaining five sites situated in suburban areas. The $CO_2$ mole fractions are highest at the MON site located in the northern part of the city of Paris, with the observed and modeled $CO_2$ of 426.2±16.9 ppm and 426.4±13.2 ppm, respectively. This is mainly related to the presence of large anthropogenic $CO_2$ emissions within the city of Paris, particularly in its north-western area (cf. Figure 1 in Lian et al. (2023)). The observed $CO_2$ mole fractions at the OBS site exhibit a more concentrated distribution, whereas the modeled $CO_2$ mixing ratios have a larger variation. Moreover,

the model tends to overestimate the mid-afternoon $CO_2$ mixing ratios at suburban stations (AND, SAC, CRE and VES) with the median plus/minus standard deviation biases varying from 0.7±6.2 ppm to 1.4±7.7 ppm.

Figure 7 shows the model-observation misfits in hourly afternoon (12-17 UTC) $CO_2$ mole fractions, averaged as a function of wind speed and direction at different stations from July 2020 to December 2022. The wind information is extracted from the WRF-Chem model at the location and sampling height at each station. The $CO_2$ data are categorized into wind classes with a bin width

of 1 m/s for wind speed and 4° for wind direction. Results for four seasons are given in Figure S7. In most suburban stations (e.g., OVS, SAC, AND and COU), the simulated $CO_2$ mole fractions are underestimated in the northeast to southeast direction, especially during the autumn and winter seasons. These model underestimations are most likely a result of issues with background $CO_2$ signals, potentially originating from either the CAMS $CO_2$ dataset or $CO_2$ emissions in remote regions. In contrast, during winter, the model tends to underestimate $CO_2$ mole fractions to a lesser degree when air flows from the cleaner Atlantic Ocean regions in

the southwest to northwest direction. In three urban stations (JUS, CAP, OBS), the variation in model-observation $CO_2$ misfit with wind direction shows a greater diversity than suburban sites. This indicates that large and heterogeneous anthropogenic $CO_2$ emissions within urban areas might counteract the underestimations caused by the background signals. Furthermore, an underestimation of the modeled $CO_2$ mole fractions was observed at the west to northwest direction at GNS station which is located 17 km north of the center of Paris. Through analysis, it has been found that this discrepancy may be attributed to the presence of a

landfill and waste treatment facility located 2.5 kilometers north of the site, while these emission sources were not included in the





emission inventory used in the model. In addition, the model-data comparison of $CO_2$ mole fractions at IGR shows a negative bias of -1.3±9.2 ppm. It may be attributed to the lower accuracy of this specific instrument (HPP6) compared to the others, as shown in Figure 5. It is noteworthy that there is also a significant underestimation of the modeled $CO_2$ mole fractions at DEF station, except for the summer period. By analyzing the relationship between spikes in $CO_2$ observations and wind patterns at DEF station,

combined with on-site investigations, this is due to local sources of contamination on the sampling rooftop, primarily originating from the direction spanning from 275° to 10° (Figure S8). This indicates that we need to carefully filter out the contaminated data at DEF for its use in the subsequent atmospheric inversion study or consider relocating the air sampling inlet at this station.

### 3.2.2 $CO_2$ spatial variations

In order to eliminate the potential errors in background $CO_2$ signals and better highlight the anthropogenic emissions in the Paris

region, we analyze the spatial variations in $CO_2$ mole fractions between pairs of stations rather than focusing only on absolute $CO_2$ values. This approach, known as the $CO_2$ gradient method, has been used in previous inversion studies for estimating $CO_2$ emissions at the city scale (Bréon et al., 2015; Staufer et al., 2016).

Figure 8 shows the distributions of the observed and modeled hourly afternoon (12-17 UTC) $CO_2$ mole fraction differences between JUS and the other stations for four seasons from July 2020 to December 2022. For CRDS sites, the median observed differences

in $CO_2$ mole fractions between the Paris city center (JUS) and suburban sites are higher than the simulated values by -0.6~2.9 ppm in winter, 1.0~1.7 ppm in spring, 0.6~1.1 ppm in summer, and 0.7~3.2 ppm in autumn. This tends to indicate that the spatial disparity in fossil fuel $CO_2$ emissions between urban and suburban areas are underestimated by 10~40% within the inventory, or that there are additional sources of $CO_2$ in the urban area which are not in the inventory (e.g., human respiration). The proximity of the HPP sites at BED, MON, CAP, IGR, to the JUS site leads to relatively small differences in $CO_2$ mole fractions. The medians

of the simulated $CO_2$ mole fraction gradients of JUS-BED and JUS-IGR are consistently larger than the observed values by (0.9, 0.8, 0.1 and 0.2 ppm) and (0.3, 0.9, 0.5, 1.9 ppm) across the four seasons (winter, spring, summer, autumn), respectively. In contrast, for JUS-MON, the modeled $CO_2$ gradients are either equal to or lower than the observations by (0, 0.4, 0.6, 1.4 ppm). The observed and modeled $CO_2$ mole fraction gradients between JUS-OBS stations exhibit significant disparities, although the sites are geographically close to each other within the Paris downtown area. Specifically, during the winter and spring seasons, the observed

median gradient values are 4.0 ppm and 3.6 ppm higher than the simulated ones, respectively, while during the summer and autumn seasons, they are lower by 1.8 ppm and 0.9 ppm. It is worth noting that the differences in $CO_2$ mole fractions among the various stations are relatively small during the summer, particularly for the HPP stations in Paris, which are situated close to JUS, with median values below 1 ppm. Given the accuracy of the HPP instrument, it is necessary to exercise caution when utilizing HPP data for atmospheric inversion via the $CO_2$ gradient method to estimate fine-scale intra-urban $CO_2$ emissions in summer, especially in

the downtown areas of Paris.

Figure 9 shows the observed (green left panels) and modeled (yellow right panels) afternoon $CO_2$ mole fraction differences between JUS and the other stations, averaged as a function of wind speed and direction from July 2020 and December 2022. A similar figure between SAC and the other stations is shown in Figure S9. Results show that the model successfully reproduces the spatial pattern of observed $CO_2$ mole fraction differences between station pairs across the urban area, which are influenced by wind fields.

As an example, for station pairs JUS-AND, JUS-VES and JUS-SAC, both the model and observations show fairly similar positive and negative values. More precisely, $CO_2$ mole fractions at the urban JUS station tend to be higher than at any suburban stations in most wind directions, resulting in positive $CO_2$ differences. However, the situation is different when suburban stations are located downwind of JUS, leading to negative $CO_2$ differences because of emissions transported at suburban stations. One obvious discrepancy between the model and observation is found at the CRE station in the 280°~330° direction. In this wind sector, the





observations show positive differences in $CO_2$ mole fractions between JUS-CRE that the model fails to capture. Further analysis reveals that this is due to an emission source from an incinerator located 2.5 kilometers northwest of the site, which is not accurately depicted in the emission inventory used in the model.

Conversely, model-observation mismatches in $CO_2$ spatial difference were noted within the city of Paris, for the station pairs JUS-MON, JUS-OBS, JUS-CDS. This is partially due to the strong heterogeneity in fossil fuel $CO_2$ emissions within an urban environment which are not well depicted by the inventory. The improvement of the inventory could be achieved through the bottom-up method, involving the collection of more detailed activity data and more accurate emission factors. Furthermore, integrating $CO_2$ observations from a dense monitoring network through the atmospheric inversion technique could also contribute to improving the inventory by correcting the spatial distribution of emissions in urban areas. This model-observation misfit could also be attributed to the complex urban structure and morphology within the central city area, such as the impact of buildings and street canyons on the energy budget and atmospheric transport. These factors lead to sub-kilometer $CO_2$ mole fraction features that cannot be captured by the WRF-Chem model at a 1-km horizontal resolution (Lian et al., 2019). This may indicate that a higher model resolution is needed to accurately represent local anthropogenic heat fluxes and small-scale processes that might affect the in-situ measurements. Nevertheless, the similarity in the $CO_2$ mole fraction difference pattern between the model and observations at station pairs such as JUS-CAP suggests that the model transport error in urban areas may not be too high.

## 4 Summary, discussion and conclusion

This study presents the development of a mid-cost instrument designed for on-site deployments and long-term continuous $CO_2$ measurements in urban environments. The effect of humidity, pressure, temperature and of $CO_2$ signal drift on the sensor performances was characterized in the lab and a calibration algorithm was implemented to adjust the raw data accordingly. Results show that correcting for the offset characterized by a daily $CO_2$ target gas injection is the most significant correction term, leading to substantial reductions in the RMSEs of $\Delta CO_2$ mole fractions between the calibrated HPP $CO_2$ data and the reference CRDS $CO_2$ measurements. The colocation evaluations have shown that the accuracies of hourly afternoon (12-17 UTC) measurements are within the range of 1.0 to 2.4 ppm among HPP instruments. The CO2calqual data processing system has been implemented for automated data quality control, calibration and storage, which makes it possible to effectively handle extensive real-time $CO_2$ measurements from the mid-cost monitoring network. It also has the capacity to process and manage data from any supplementary stations in the future.

Field measurements conducted over the last 2.5 years show that the mid-cost HPP instruments do not have the same level of stability as the CRDS instruments when used for long-term continuous outdoor measurements. Therefore, the automatic detection of data loss, swift identification of its causes, and efficient instrument maintenance become important when operating a dense mid-cost $CO_2$ monitoring network. Operations require maintenance of the HPP instrument, including a replacement of the target tank every 4-5 months. As part of this maintenance, the HPP is positioned on the LSCE rooftop for at least 3 days, for an evaluation of its performance with comparison to a reference CRDS instrument. The main objective is to verify the HPP performance with this on-going on-site tank. Current results show that since the first installation in July 2020 up to the present, all 8 HPP instruments have remained in operation without significant performance degradation. Our recent colocation evaluations indicate that the potential measurement bias due to the gradual loss of $CO_2$ sensitivity of the sensor over time has been effectively corrected by the target tank. Consequently, there is presently no need for additional sensitivity tests in the laboratory to update the various calibration coefficients.





It should be pointed out that $CO_{2_{offset}}$ in our current HPP data calibration relies fully on the daily target gas correction. There is a potential risk associated with this method, as a tank issue could lead to notable measurement bias. We should consider applying the default $CO_2$ offsets determined during the initial lab calibration for each HPP sensor, and then use the daily target gas measurement to correct the measurement offset through an additive adjustment. However, different HPP instruments exhibit

varying degrees of measurement drifts during their prolonged outdoor operations. Consequently, the effectiveness of the default $CO_2$ offsets in providing corrections also changes accordingly. Our recent analyses (Figure S10) indicate that in the case of a more stable HPP instrument like HPP7 at OBS, the default $CO_2$ offset could consistently contribute effectively to calibration, resulting in a reduced correction requirement from the target gas. Conversely, for an HPP instrument undergoing gradual slow drifts like HPP8 at BED, the corrective impact of the default $CO_2$ offset weakens, leading to an increased reliance on the correction of the

target gas. Field measurements at these HPP stations are being continued to monitor the instrument performance over their operational lifespan. Meanwhile, we also need to consider how to handle the biased observed data if the HPP measurements have a noticeable drift in comparison to the reference CRDS instrument during this colocation evaluation. Such drift may induce some non-linear or time-varying impacts on the measured $CO_2$ mole fractions as a function of continuous operations over the past weeks or months.

The model-observation comparisons show that HPP instruments exhibit similar magnitudes of model-observation misfits in $CO_2$ mole fraction when compared to CRDS instruments. The WRF-Chem model at 1-km horizontal resolution reproduces the observed cross-city $CO_2$ mole fraction differences among both HPP and CRDS station pairs. The analysis of $CO_2$ spatial gradients indicates that observations from both HPP and CRDS instruments can help identify potential underestimations of $CO_2$ emissions and the absence of emission point sources in the inventory in the vicinity of some stations. Considering also that both the local $CO_2$ signals

and the uncertainty in fossil fuel $CO_2$ emissions from different inventories exhibit considerably larger magnitudes than the accuracies of HPP instruments, the HPP data have promising potential in providing valuable insights into $CO_2$ variations in and around the city of Paris. This makes them suitable for application in subsequent inverse modeling endeavors to improve the spatial representation of $CO_2$ emissions from the inventory. However, afternoon $CO_2$ mole fraction differences between station pairs in summer, especially the HPP stations located within the Paris city limits, are quite small, typically below 1 ppm. In these cases, the

accuracy of the HPP instruments is not sufficient to identify model-observation misfits that would be generated by an error in the emission estimate in the downtown areas of Paris. Furthermore, this study mainly focuses on $CO_2$ observations during the afternoon as they are commonly used for atmospheric inversions. The accuracies of hourly HPP measurements in non-afternoon hours (18-11 UTC) are slightly worse than those observed in the afternoon, with RMSEs ranging from 1.3 to 3.9 ppm among HPP instruments. Taking also into account the large errors associated with atmospheric transport models during nighttime, the assimilation of

nighttime $CO_2$ data from both CRDS and HPP instruments into the inversion system appears impractical at this stage.

Currently, $CO_2$ monitoring instruments in Paris are placed on rooftops or towers to increase the spatial representativeness of the measurements. It is noteworthy that the deployment strategy of these mid-cost medium-precision instruments can be adjusted based on the diverse objectives of $CO_2$ emission monitoring in urban areas. For instance, strategically deploying instruments in close proximity to anthropogenic sources such as buildings and traffic can substantially improve the signal-to-noise ratio, enabling more

accurate monitoring of different $CO_2$ emission sources within the city. This study demonstrates that there is a continued need to filter out locally contaminated observation data, even after implementing a spike detection algorithm (Eq. 2) at the DEF site. Indeed, we must find a suitable approach for spike detection and data filtering that will be used in an urban atmospheric inversion system. It is essential to determine the scale we are targeting and understand the criteria for distinguishing between good and bad "local" distances, along with the corresponding sizes of spikes.



The 2.5-year experience in using these 8 mid-cost medium-precision instruments also provides insights for the development of the next iteration of these instruments. A further deployment of a dense atmospheric observation network, a high-resolution transport modeling and a spatially explicit inversion system would allow to solve for the spatial distribution of urban $CO_2$ emissions at the grid scale finer than 1 km resolution. In 2021, the ICOS Cities, also referred to as the PAUL project (Pilot Application in Urban

Landscapes - towards integrated city observatories for greenhouse gases, https://www.icos-cp.eu/projects/icos-cities), was launched as part of the European Union's Horizon 2020 research and innovation program. Its primary objective is to establish integrated city observatories for greenhouse gases and focuses on assessing various measurement techniques to determine fossil fuel emissions in relation to carbon dioxide levels in the atmosphere. To achieve this, the project has constructed testbeds in three cities of varying sizes: Paris, Munich, and Zürich. As part of this initiative, a plan has been set to install additional two CRDS and

21 mid-cost instruments in Paris starting from the year 2023 to further enhance the $CO_2$ monitoring capabilities, enabling to gain a comprehensive understanding of $CO_2$ emissions in urban environments.

**Author contribution**

JL and OL conceptualized this study. OL and LL developed and integrated the HPP instruments. OL, LL and MC designed and performed the sensor laboratory sensitivity tests and the initial data calibrations. OL, MC, LL, KC and LM contributed to the field

deployments. HU, OL, KC, LL and JL were involved in the development of the CO2calqual data processing system. JL configured and ran the WRF-Chem model simulations, and conducted all the analyses for the manuscript. JL, OL, MR, HU, TL, FMB, GB and PC collaborated on interpretation of the results. JL wrote the manuscript with contributions and suggestions from all authors.

**Code/Data availability**

The $CO_2$ observations at 7 CRDS stations are available on request from Michel Ramonet (michel.ramonet@lsce.ipsl.fr).

The $CO_2$ observations at 8 HPP stations are available on request from Jinghui Lian (jinghui.lian@suez.com) and Hervé Utard (herve.utard@origins.earth).
The Origins.earth $CO_2$ inventories are available on request from Hervé Utard (herve.utard@origins.earth).

**Competing interests**

The authors have no competing interests to declare.

**Acknowledgements**

The authors would like to thank SUEZ Group, Ville de Paris and ICOS Cities for the support of this study. We would like to thank SUEZ Group for supporting the deployment of mid-cost instruments at VES and CRE sites. Thanks to Covivio, Cité de l'architecture et du Patrimoine, Eau de Paris, Ville de Paris, Institut Gustave Roussy, Observatoire de Paris for granting access to the DEF, CAP, MON, BED, IGR and OBS sites, respectively. Thanks to Allianz and BNP Paribas for their support in deploying

the Citylights site. Thanks also to Enviroearth, our contractors, for their amazing work in installing the HPP instruments and providing continuing support. We would like to acknowledge our former colleagues (Thibault Vignon and Danielle Bengono) at Origins.earth and colleagues (Nathanael Laporte, Guillaume Nief and Tanguy Martinez) from LSCE for their assistance in maintaining the sensors and their contributions to data transfer, calibration and analysis. We thank Xiaobo Yang and Anna Agusti-

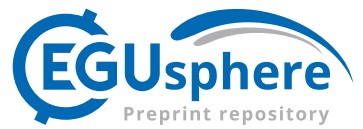

Panareda from ECMWF for providing the near-real-time (NRT) CAMS $CO_2$ dataset. We thank Rateb Sayah and Philippe Wlodkowski from the digital team at SUEZ Consulting for their ongoing technical assistance.

**Financial support**

This research has been funded by SUEZ Group together with LSCE under the collaboration convention "Pour développer un outil d'estimation des émissions de $CO_2$ sur Paris et les communes avoisinantes". The authors have also received funding from the ICOS Cities, aka Pilot Applications in Urban Landscapes - Towards integrated city observatories for greenhouse gases (PAUL) project by the European Union's Horizon 2020 research and innovation program under grant agreement NO.101037319.

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



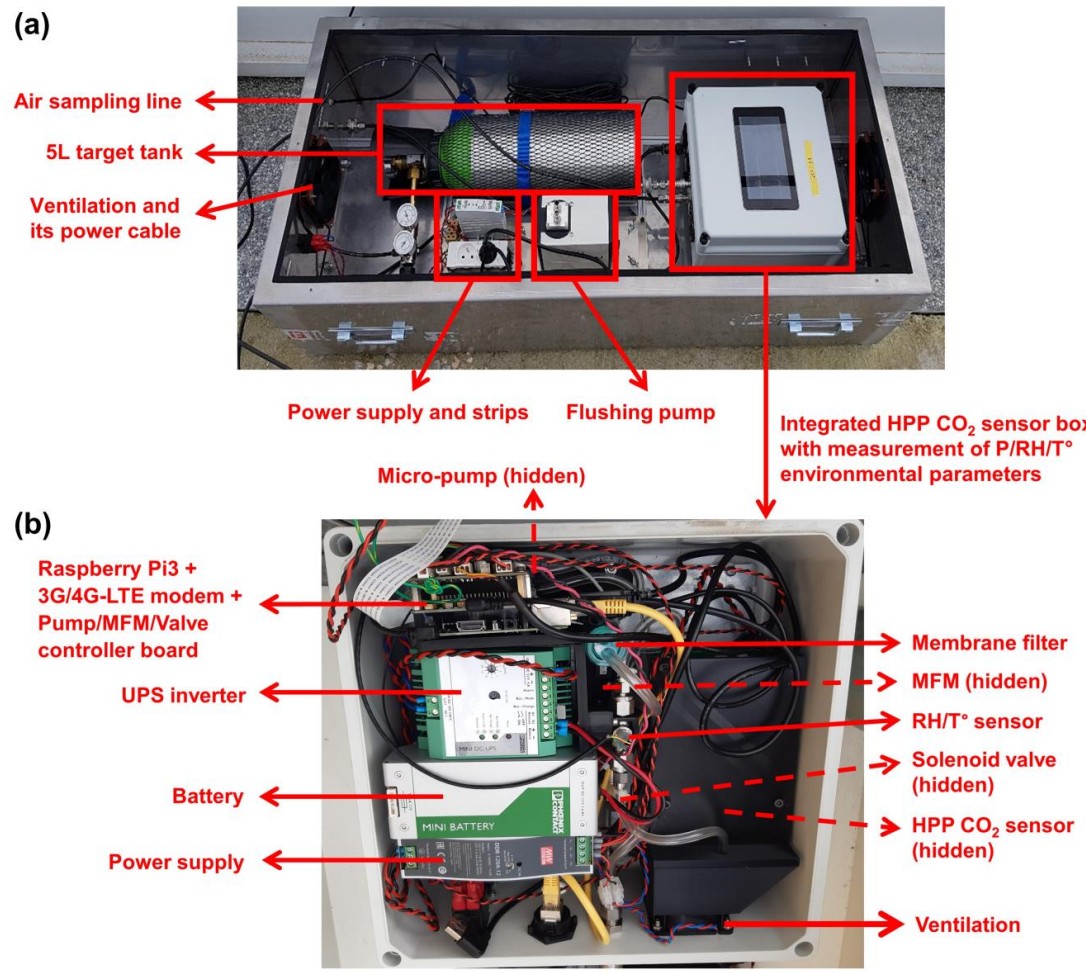

**Figure 1: Components of the integrated mid-cost CO₂ measuring instrument.**



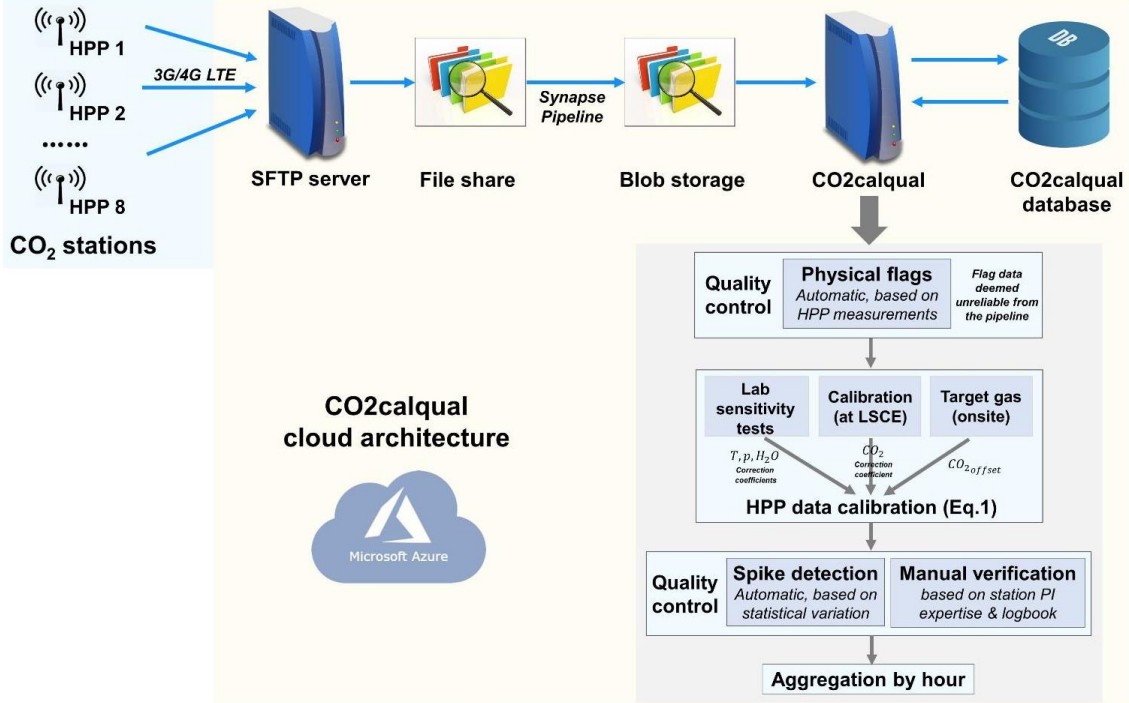

**Figure 2: Schematic of the data processing architecture for the HPP monitoring network.**



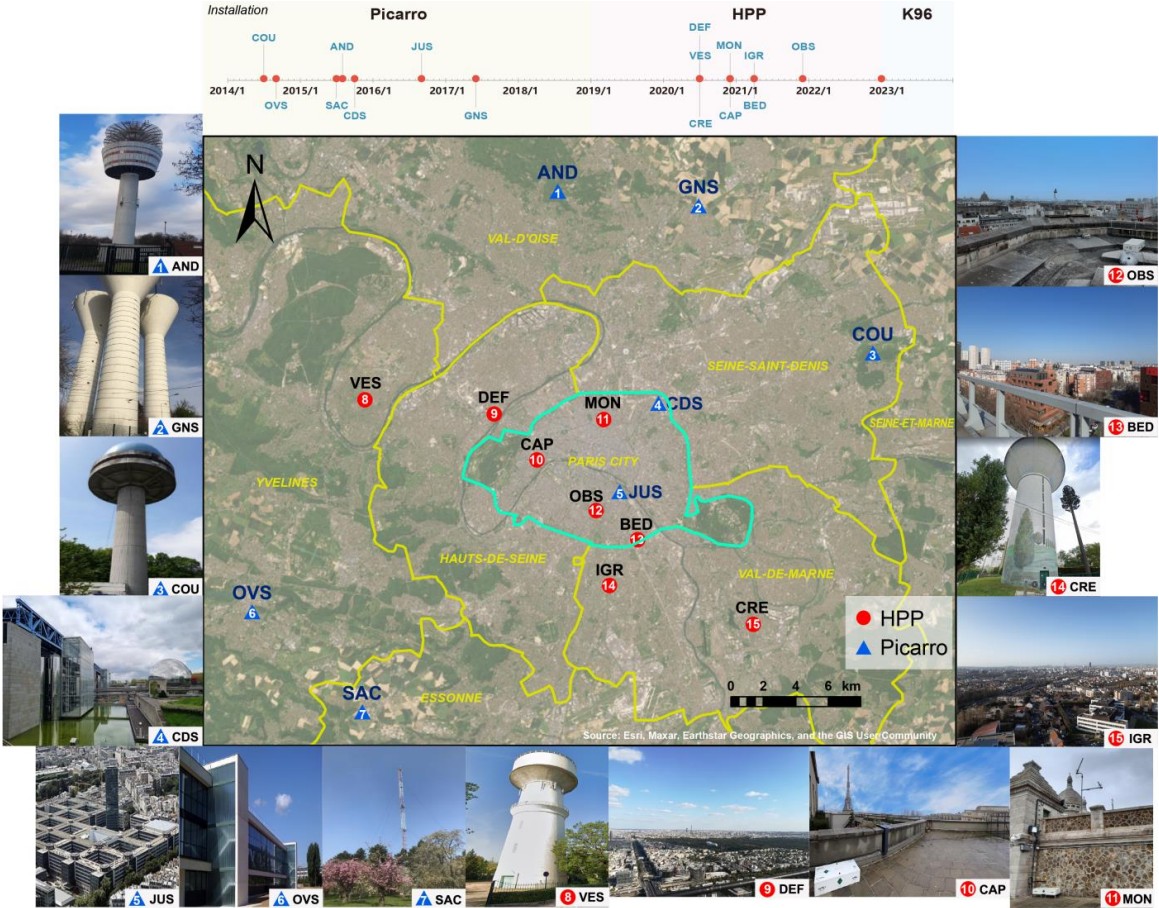

**Figure 3: Locations and photos of the eight HPP and seven CRDS Picarro $CO_2$ measurement stations in the Paris region which includes the city of Paris (cyan line) and its surrounding seven departments (yellow lines). The installation dates of the sensors are shown in the top panel. Image credits: JUS ©LOIC VENANCE/AFP. CDS ©BRUNO URBANI from google map. VES and CRE © google map. AND https://rncmobile.net/site/10762, last access Jan 25th 2024. OVS https://www.ovsq.uvsq.fr/en, last access Jan 25th 2024.**





**Figure 4: Time series of hourly CO₂ mole fractions measured by each HPP sensor and the collocated reference CRDS measurements at LSCE laboratory rooftop during a colocation period of 3~11 days. The tables show the RMSE values of hourly ΔCO₂ mole fractions between HPP and CRDS in terms of each correction component ($H_2O, T, p, IC_1$ and $CO_{2_{offset}}$) in Eq. (1), for both the afternoon period (12-17 UTC) and the entire day. Note that corrections are cumulative from left to right. The light grey shaded areas indicate the injection of target gases in the middle of the day.**





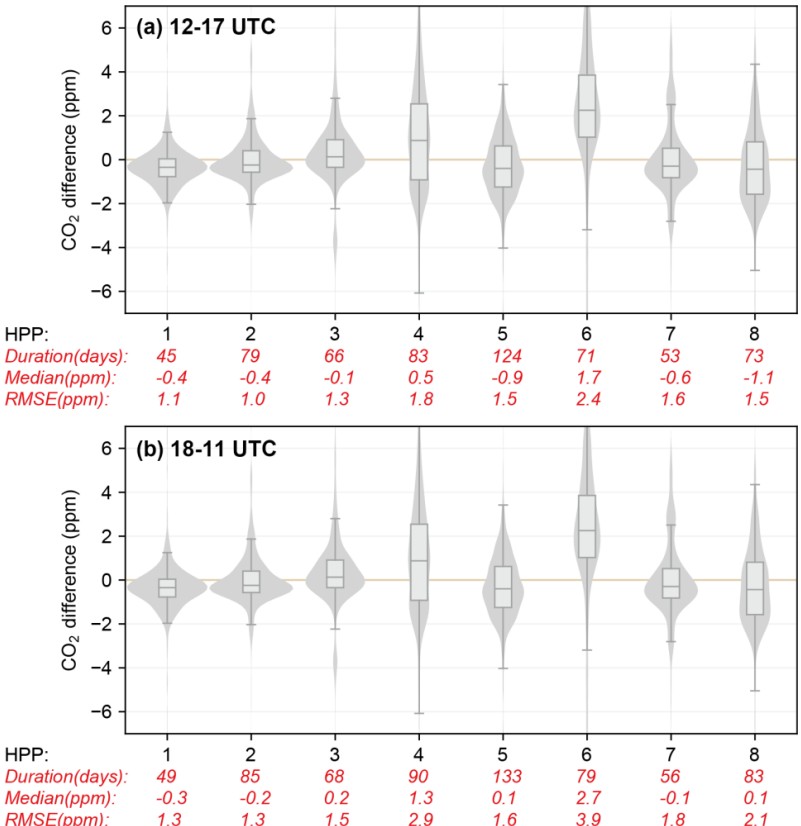

**Figure 5: Differences (median and RMSE) in hourly CO₂ mole fractions in the (a) afternoon (12-17 UTC) and (b) other times of the day (18-11 UTC) between the calibrated HPP data and the CRDS measurements during all the intercomparison periods. The midpoint, the box and the whiskers represent the 0.5 quantile, 0.25/0.75 quantiles, and 0.1/0.9 quantiles respectively.**

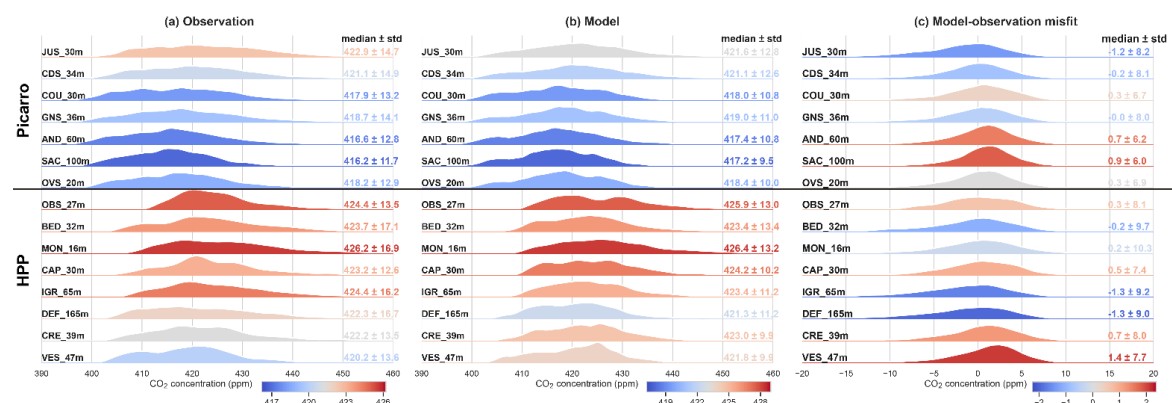

**Figure 6: Distributions of (a) observed and (b) modeled hourly afternoon (12-17 UTC) CO₂ mole fractions, as well as the (c) model-observation misfits at 7 CRDS Picarro and 8 HPP stations, respectively over the period of July 2020 to December 2022. Both CRDS and HPP stations are displayed in a top-to-bottom sequence, corresponding to their increasing distance from the JUS station. The station names are given together with their respective sampling heights above ground level. The median (shown also in colorbar) and standard deviation of CO₂ mole fractions at each station are shown on the right.**



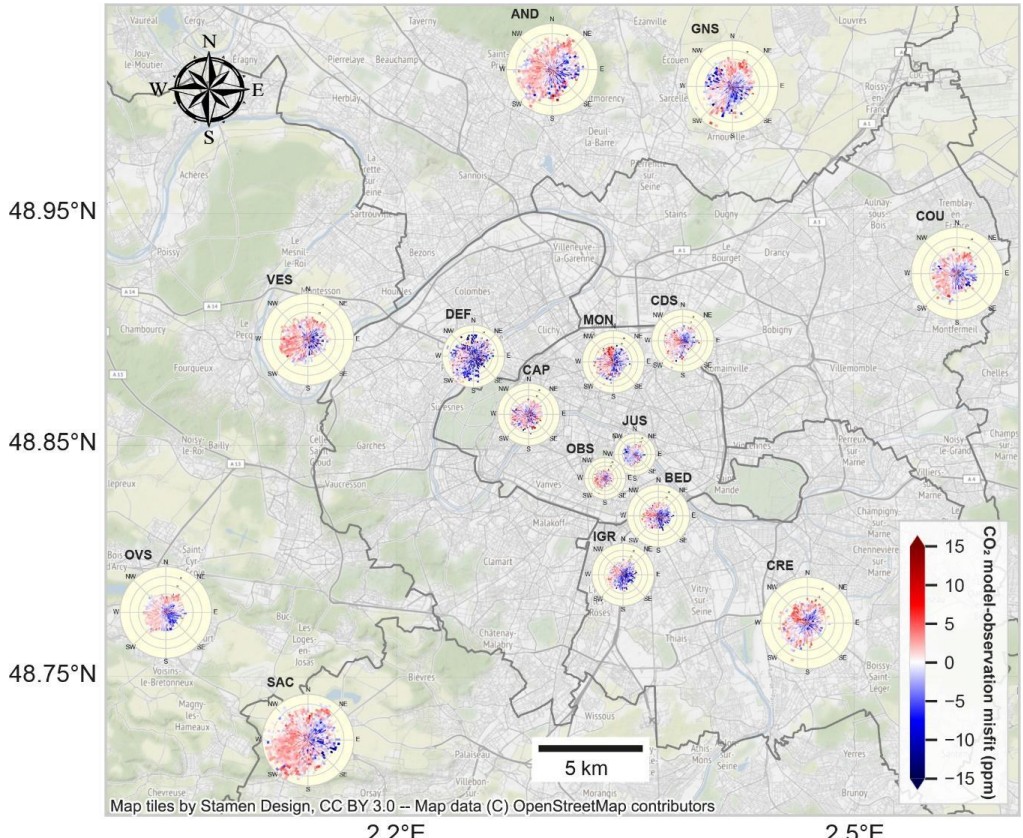

**Figure 7: Model-observation misfits in hourly afternoon (12-17 UTC) CO₂ mole fractions, averaged accounting for wind speed and direction at 7 CRDS and 8 HPP stations over the period of July 2020 to December 2022. The different sizes of the polar panels hold no specific meaning and are merely adjusted to avoid overlaps.**





**Figure 8. Distributions of the observed and modeled hourly afternoon (12-17 UTC) CO₂ mole fraction differences between JUS and the other stations for four seasons from July 2020 to December 2022. The red solid lines and numbers represent the median values. The dash grey lines represent the first and third quantiles. The distances from each site to the JUS site (in kilometers) are provided on the x-labels.**



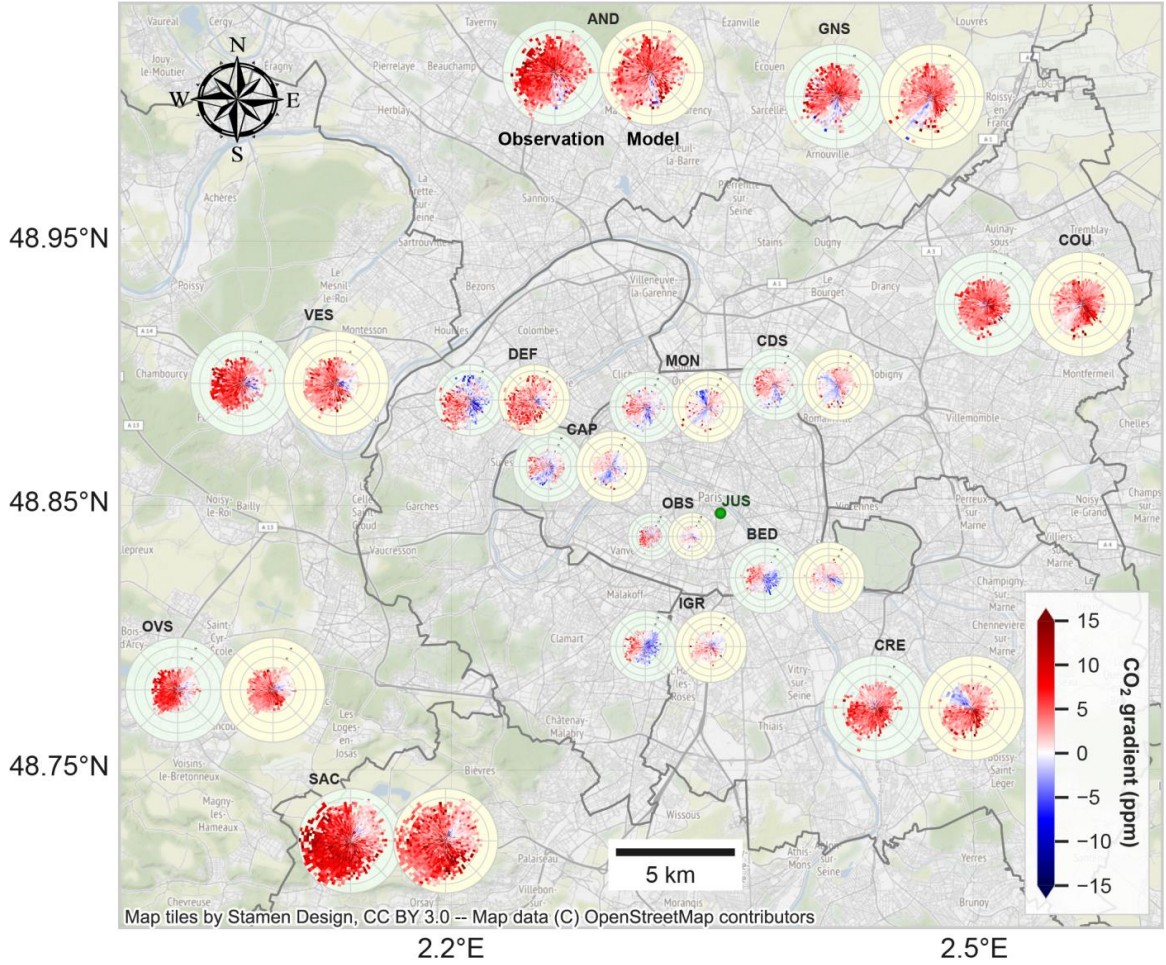

**Figure 9.** Observed (green panel) and modeled (yellow panel) CO₂ mole fraction differences between JUS and all the other stations, averaged accounting for wind speed and direction over the period of July 2020 to December 2022. Only the afternoon (12-17 UTC) data are used. The different sizes of the polar panels hold no specific meaning and are merely adjusted to avoid overlaps.



**Table 1. Summary of the laboratory tests**

|  | Purpose | Location | CRDS as a reference | Air measured | Range of $T$, $p$ and $H_2O$ | Range of $CO_2$ (ppm) | Duration |
|---|---|---|---|---|---|---|---|
| Water vapor | Correlation between $CO_2$ and $H_2O$ | Laboratory | No | Target cylinder | 0 - 2.5% v | ~420 ppm | 3 hours |
| Pressure | Correlation between $CO_2$ and $p$ | Climatic chamber | Yes | Indoor air in climatic chamber | 800 - 975hPa | ~450 ppm | 15 hours |
| Temperature | Correlation between $CO_2$ and $T$ | Climatic chamber | Yes | Indoor air in climatic chamber | -10 - +40°C | ~450 ppm | 15 hours |
| Calibration | Setup calibration equation | Laboratory | Yes | Calibration cylinders | +22°C Atmospheric pressure | 400 - 600 ppm | 70 minutes |
| Colocation | Evaluation of HPP performance in outdoor conditions | Laboratory rooftop | Yes | Ambient air | -6.4 - +35.3°C 960 - 1024hPa | 402 - 535 ppm | At least two weeks. Varies by case |

**Table 2. Information about the eight HPP and seven CRDS Picarro $CO_2$ measurement stations**

|  | Site | Acronym | Latitude (°) | Longitude (°) | Height AGL (m) |
|---|---|---|---|---|---|
| HPP | Le Vésinet | VES | 48.8960 | 2.1415 | 47 |
|  | La Défense | DEF | 48.8892 | 2.2506 | 165 |
|  | CAPA | CAP | 48.8632 | 2.2908 | 30 |
|  | Montmartre | MON | 48.8863 | 2.3421 | 16 |
|  | Observatoire de Paris | OBS | 48.8364 | 2.3367 | 27 |
|  | Bédier | BED | 48.8197 | 2.3714 | 32 |
|  | Créteil | CRE | 48.7733 | 2.4693 | 39 |
|  | Institut Gustave Roussy | IGR | 48.7942 | 2.3481 | 65 |
| Picarro | Jussieu | JUS | 48.8464 | 2.3561 | 30 |
|  | Cité des Sciences | CDS | 48.8956 | 2.3880 | 34 |
|  | Andilly | AND | 49.0126 | 2.3018 | 60 |
|  | Coubron | COU | 48.9242 | 2.5680 | 30 |
|  | Gonesse | GNS | 49.0052 | 2.4205 | 36 |
|  | OVSQ | OVS | 48.7779 | 2.0486 | 20 |
|  | Saclay | SAC | 48.7227 | 2.1423 | 15, 60 and 100 |

**Table 3. Summary of correction coefficients derived from the sensitivity tests for each HPP sensor**

|  |  | $H_2O$ (ppm/%v) | | | $p$ (ppm/hPa) | | $T$ (ppm/°C) | | | $CO_2$ | |
|---|---|---|---|---|---|---|---|---|---|---|---|
|  |  | $IH_1$ | $IH_2$ | $R^2$ | $IP_1$ | $R^2$ | $IT_1$ | $IT_2$ | $R^2$ | $IC_1$ | $R^2$ |
| HPP1 | VES | -2.40E-03 | -0.70E-03 | 0.986 | 0.057 | 0.999 | -1.746 | 0.015 | 0.968 | 1.045 | 1 |
| HPP2 | CRE | -1.94E-03 | -1.71E-03 | 0.997 | 0.058 | 0.999 | -0.673 | 0.003 | 0.979 | 1.075 | 1 |
| HPP3 | DEF | -1.64E-03 | -1.29E-03 | 0.991 | 0.065 | 0.998 | -0.760 | 0.001 | 0.972 | 1.042 | 1 |
| HPP4 | CAP | -3.92E-03 | -0.48E-03 | 0.998 | 0.059 | 0.999 | -2.010 | 0.013 | 0.991 | 0.997 | 1 |
| HPP5 | MON | -0.36E-03 | -1.40E-03 | 0.990 | 0.060 | 0.999 | -0.446 | 0.003 | 0.934 | 1.038 | 1 |
| HPP6 | IGR | -3.96E-05 | -2.18E-03 | 0.934 | 0.060 | 0.998 | -5.692 | 0.062 | 0.995 | 1.038 | 1 |
| HPP7 | OBS | -0.18E-03 | -1.98E-03 | 0.988 | 0.055 | 0.999 | 1.144 | -0.015 | 0.804 | 1.073 | 1 |
| HPP8 | BED | 0.75E-03 | -1.64E-03 | 0.850 | 0.060 | 0.999 | -1.620 | 0.013 | 0.983 | 1.053 | 1 |