# Peer review of "Development and deployment of a mid-cost CO2 sensor monitoring network to support atmospheric inverse modeling for quantifying urban CO2 emissions in Paris"

_EGUsphere, 2024_

## Author Comment (AC1)

We would like to thank the reviewer for the comments and suggestions to our manuscript. In the following, we answer to the reviewer's comments and indicate the changes in the manuscript that were implemented according to the recommendations. The comments are in black. Our answers are in blue.

**Referee #1:**

The manuscript titled "Development and deployment of a mid-cost $CO_2$ sensor monitoring network to support atmospheric inverse modeling for quantifying urban $CO_2$ emissions in Paris" is generally well written and scientifically justified. We recommend it to be published after addressing the following points.

It would be helpful to have additional discussion of the spatial density of the network. The reader can deduce this from Figure 1, but average distance to the nearest site would be helpful to include in Section 2.4. How does the spatial density of the HPP network compare to the density of the Picarro network? How does the HPP spatial density compare to the previous studies of similar moderate-cost sensor networks (i.e. Wu et al., 2016; Turner et al., 2016)?

**Response:**

We thank the referee for the positive comments on our manuscript and the valuable suggestion concerning the spatial density of the network. The following figure shows the distances between sites in kilometers for the real urban $CO_2$ monitoring network containing 15 sites in this study. The average distance to the nearest site in the CRDS Picarro network is 8.7km, while for the HPP network, it is 4.9 km. In the combined CRDS Picarro and HPP network, this average distance reduces to 6.1 km. Both Wu et al. (2016) and Turner et al. (2016) conducted observing system simulation experiments (OSSEs) using pseudo-measurement networks. Wu et al. (2016) evaluated the potential of inversion with networks consisting of 10, 30, 50, or 70 stations. In Turner et al. (2016), the pseudo-measurement network included 34 sites at roughly 2 km spacing covering an area of roughly 400 km$^2$. Therefore, the real HPP network with 8 sites used in this study has a lower spatial density compared to the pseudo-measurement networks analyzed in the two previous OSSE studies.

We have added the following sentences in section 2.4 and Figure S5 in the supplement:

"Figure S5 shows the distances between stations in kilometers for the $CO_2$ monitoring network in this study. The average distance to the nearest site in the CRDS network is 8.7km, while for the HPP network, it is 4.9 km. In the combined CRDS and HPP network, this average distance reduces to 6.1 km."

[Figure]

| Site (km) | JUS | CDS | COU | GNS | AND | SAC | OVS | OBS | BED | MON | CAP | IGR | DEF | CRE | VES | Average |
|---|---|---|---|---|---|---|---|---|---|---|---|---|---|---|---|---|
| Picarro | 5.9 | 5.9 | 13.6 | 8.7 | 8.7 | 9.2 | 9.2 | | | | | | | | | 8.7 |
| HPP | | | | | | | | 3.2 | 3.2 | 4.6 | 4.1 | 3.3 | 4.1 | 8.9 | 8 | 4.9 |
| Picarro +HPP | 1.8 | 3.5 | 13.6 | 8.7 | 8.7 | 9.2 | 9.2 | 1.8 | 3.2 | 3.5 | 4.1 | 3.3 | 4.1 | 8.9 | 8 | 6.1 |

Figure S5. Site-to-site distance in kilometers. The distances to the nearest site for each site are highlighted in bold black font (read by rows) and are summarized in the table.

P and p are used seemingly interchangeably for pressure. (Eq 1 uses P, but p is used frequently in the text and other figures).

**Response:**

Modified. We have used P consistently in the text, equation, figure and table.

Figure 1 and Figure 2 emphasize the hardware of the sensor and the data infrastructure. These are important to include but might be better to be included in the SI than the main text as the figures contain a lot of details that are not necessarily central to the main points of the paper.

**Response:**

The photos and flowcharts in Figure 1 and Figure 2 are closely linked with Sections 2.1, 2.2, and 2.3 of the main text. They provide a visual representation of the assembly components and calibration process for the mid-cost instrument. Therefore, we prefer to keep them in the main manuscript as they could offer additional clarity and detail to complement the textual description of the instrument development. These visual demonstrations may facilitate readers in better understanding the information presented in the text.

Figure 6 is too small to read comfortably, especially the text.

**Response:**

Following the suggestion from Reviewer #3, panel (b) with the modeling data was omitted, and the aspect ratio of the figure was adjusted. We also bolded the font for an improved reader

experience.

Figure 8: Add some break / distinction between the Picarros and HPPs

**Response:** Added as suggested.

Figure 9: Please consider flipping the direction of the difference (site - JUS), so that lower than JUS values are negative. Currently this figure gives the impression of elevated concentrations outside of the urban center. Alternatively, clarify in the figure caption that the difference is JUS minus other sites. Same note for figure S9.

**Response:**

As suggested, we have clarified it both in the figure captions and main texts. The current order of the subtraction is intended to show that $CO_2$ concentrations at urban sites are generally higher than those at suburban sites. Note that Figure S9 has become Figure S12 in the revised manuscript.

Figure 9: "The $CO_2$ differences are calculated as JUS minus the other stations."

Figure S12: "The $CO_2$ differences are calculated as the other stations minus SAC."

Main text: "Figure 9 shows the observed (green left panels) and modeled (yellow right panels) afternoon $CO_2$ mole fraction differences between JUS and the other stations, averaged as a function of wind speed and direction from July 2020 and December 2022. The $CO_2$ differences are calculated as JUS minus the other stations. Additionally, Figure S12 presents a similar comparison, but with $CO_2$ differences of other stations minus SAC."

Page 9, line 9: Could the author(s) comment or speculate on the variable sensor performance? Is this a difference in sensor hardware or experimental condition?

**Response:**

There are many reasons for the varying performance of HPP instruments, making it difficult to specifically determine and explain the exact causes. For instance:

Firstly, as for the HPP sensor itself, although they are the same model, minor differences in the sensor production process, including the quality of raw materials (e.g., electronic components), manufacturing, quality control, and so forth, may lead to variations in the performance and stability of the sensors.

Secondly, when integrating the HPP sensor with components such as the solenoid valve, micro-pump, membrane filter, and battery into the HPP measurement unit, even with the same design, minor variations in manual production and operational techniques can result in slight changes during the production process. These variations may also lead to differences in the performance of the instruments.

Thirdly, during the sensor laboratory tests presented in section 2.1, different HPP sensors exhibit varying fits for temperature and $H_2O$, as well as different residual errors (see Figure S3 and Table 3). This is also among the reasons why different HPP instruments have varying accuracy.

Page 1, line 14: ambiguous pronoun reference (Its)

**Response:** Change from "Its measurements" to "These dense measurements".

Page 1, line 16: Missing word (should be "to separate")

**Response:** Corrected as suggested.

Page 1, line 28: should be "prospects for"

**Response:** Corrected as suggested.

Page 1, line 33: "spatial and temporal variations" of what? Emissions?

**Response:** Change to "spatial and temporal variations in emissions".

Page 3, line 15: "in dimensions" is redundant

**Response:** Deleted.

Page 3, line 21: a SHT75 -> an SHT75

**Response:** Corrected.

Page 12, line 33 "on-going" -> "ongoing"

**Response:** Corrected.

---

## Author Comment (AC2)

We would like to thank the reviewer for the comments and suggestions to our manuscript. In the following, we answer to the reviewer's comments and indicate the changes in the manuscript that were implemented according to the recommendations. The comments are in black. Our answers are in blue.

**Referee #2:**

The manuscript, "Development and deployment of a mid-cost $CO_2$ sensor monitoring network to support atmospheric inverse modeling for quantifying urban $CO_2$ emissions in Paris," provides a thorough description of the development and evaluation of a new urban mid-cost sensor monitoring network. This is a well-written document that is useful for those working on greenhouse gas quantification with urban monitoring networks.

**Response:**

We thank the referee for the positive comments on our manuscript.

The authors should address the following (minor) comments:

Page 4, lines 1-2: The wording is a bit unclear - Is the flushing pump installed or is it not? The text says it "could be" installed, but Figure 1a shows that it is installed.

**Response:**

The installation of a flushing pump depends on the specific requirements of each site. If the site configuration necessitates a long sampling line, then a flushing pump will be installed upstream of the integrated $CO_2$ box to boost the flow rate, thereby reducing the residence time in the sampling system. Conversely, for sites that do not require a long sampling line, the installation of a flushing pump is unnecessary.

We have revised the sentence to better clarify this point:

"Optionally, a flushing pump (Figure 1a) could be installed upstream the integrated $CO_2$ box in order to increase the flow rate and thus decrease the residence time in the sampling system. The necessity of installing this pump depends on the specific conditions at the measurement site. Sites with a long sampling line (EATON Synflex 1300) would benefit from its use, whereas a short line may not need it."

Page 5: In lines 35-37, the authors describe the target gas calibration using only 2 minutes to allow for flushing, but the calibration in parallel with the CRDS instrument requires 7 minutes to flush as described in lines 16-17. Could the authors provide justification for the shorter flushing time?

**Response:**

We have added the following paragraph in the supplement (Text S1 and Figure S4) to explain the settings for these two flushing times:

"To mitigate delays in sensor responses and ensure stability, thorough $CO_2$ flushing of the sensor cell is necessary. During the $CO_2$ correction coefficient $IC_1$ determination process, we sequentially sampled $CO_2$ mole fraction for a duration of 10 minutes, with 7 minutes dedicated to flushing and only the last 3 minutes of data used. During the on-site daily target gas injection for the $CO_{2offset}$ calculation, we sampled $CO_2$ mole fraction for a duration of 3 minutes, with 2 minutes of flushing and only the last minute of data used.

The differences in flushing times are due to two reasons. First, the $CO_2$ correction coefficient $IC_1$ is determined through a multipoint $CO_2$ regression using the seven mole fraction values

assigned within the 400-600 ppm range. Conversely, the $CO_2$ concentration in the target tank (which contains dry compressed natural air, pressurized at 200 bars and calibrated in $CO_2$) is supposed to be close to the ambient air $CO_2$ concentration on-site during midday. The step between two different $CO_2$ concentrations in the $IC_1$ determination process is greater than that during the target tank injection for drift correction, thus requiring a longer flushing time to achieve stabilization. Second, the CRDS and the mid-cost HPP sensor do not measure at the same flow rate, approximately 0.25 LPM for the CRDS and about 1 LPM for the HPP. They also have different precision targets. The CRDS sensor requires an extended period of target gas measurements to achieve a stability of less than 0.05 ppm, which is suitable for applications beyond this specific intercomparison. Therefore, the flushing time in the $IC_1$ determination process, when the HPP sensor measures in parallel with the CRDS, is expected to be longer.

Before implementing this setting, we carried out several sensitivity tests on the sensor performance with a daily injection of target gas lasting 5 minutes at LSCE laboratory. The following figure shows the evolution of target gas injection duration in relation to the differences in $CO_2$ concentration between the other 4 minutes and the 3[rd] minute at one HPP sensor (HPP3) over 26 days. It demonstrates that a 3-minute target gas injection, specifically utilizing the 3[rd] minute data, proved to be sufficient. The added value of the 4[th]- and 5[th]- minute injection is rather limited. Therefore, the choice of a two-minute flush serves as a good compromise between maintaining good sensor performance (ensuring a target accuracy of 1 ppm) and minimizing gas consumption (thereby extending the lifespan of the tank and reducing associated maintenance requirements)."

[Figure]

Figure S4. The evolution of target gas injection duration in relation to the differences in $CO_2$ concentration between the other 4 minutes and the 3[rd] minute at one HPP sensor (HPP3), with a daily injection of target gas lasting 5 minutes over 26 days at LSCE.

Page 7, lines 15-16: It is unclear why the authors chose to apply one-minute averaging to the data before calibration.

**Response:**

The HPP instruments sample data approximately every second, resulting in a large amount of data over long periods. To facilitate subsequent data storage, processing, and retrieval, we first average the second-level data into minute-level data. Theoretically, if the amount of data collected per minute is consistent, averaging the second-level data directly into hourly data should yield the same result as first averaging it into minute-level data and then into hourly data. However, due to potential variations in the actual data collection amount, there might be

very slight differences between these two processes. Additionally, since our model outputs data on an hourly basis, the minute-level observation data remains sufficient for comparison with the model. We have revised the sentence to clarify this point:

"These data are averaged at the temporal resolution of one minute to simplify data storage, processing, and retrieval. Following this, a calibration procedure is applied to the one-minute data."

Page 11, lines 18-19: I do not agree with the authors when they state with certainty that the small differences in $CO_2$ mole fraction between these sites and JUS can be attributed solely to their proximity to the site when OBS, the site closest to JUS, has a substantial difference in mole fraction. I would expect the city center to have substantial heterogeneity in $CO_2$ mole fraction.

**Response:**

We agree with the reviewer that the city center is expected to have substantial heterogeneity in $CO_2$ mole fraction. In fact, the original sentence is inaccurately expressed, which causes a misunderstanding. What we intended to say is that the magnitude in $CO_2$ mole fraction differences between these urban sites and the urban JUS site are smaller than those between JUS and the suburban sites. We have rephrased the sentence as follows:

"The proximity of the HPP urban sites at BED, MON, CAP, IGR, to the JUS site leads to relatively smaller differences in $CO_2$ mole fractions, compared to those between JUS and the suburban sites."

Figure 9 and paragraph beginning page 11, line 31: I recommend the authors specify the sign convention of the gradient.

**Response:**

As suggested, we have clarified it both in the figure captions and main texts. The current order of the subtraction is intended to show that $CO_2$ concentrations at urban sites are generally higher than those at suburban sites. Note that Figure S9 has become Figure S12 in the revised manuscript.

Figure 9: "The $CO_2$ differences are calculated as JUS minus the other stations."

Figure S12: "The $CO_2$ differences are calculated as the other stations minus SAC."

Main text: "Figure 9 shows the observed (green left panels) and modeled (yellow right panels) afternoon $CO_2$ mole fraction differences between JUS and the other stations, averaged as a function of wind speed and direction from July 2020 and December 2022. The $CO_2$ differences are calculated as JUS minus the other stations. Additionally, Figure S12 presents a similar comparison, but with $CO_2$ differences of other stations minus SAC."

Technical comments:

Page 1, line 14: "Its" should be "Their" if the authors are referring to the mid-cost sensors.

**Response:** Change from "Its measurements" to "These dense measurements".

Figure 4: The tables in each figure should have units.

**Response:** The units are added as suggested.

Page 8, line 34: The authors are missing an "and" here, and I have a suspicion that they meant to use HPP5 rather than HPP7 as an example for improvement from the p correction.

**Response:** We have revised this sentence, as well as the two preceding and following sentences, to improve the accuracy of expression.

Figure 6: The text is hard to read because of the small font size and light colors.

**Response:** Following the suggestion from Reviewer #3, panel (b) with the modeling data was omitted, and the aspect ratio of the figure was adjusted. We also bolded the font for an improved reader experience.

Figures 7 and 9: The wind roses for OBS (both figures) and JUS (Figure 7) are too small to interpret.

**Response:** The sizes of the wind roses for OBS and JUS were adjusted to avoid overlaps while also trying to keep them as close as possible to their actual spatial locations in the map. We were concerned that enlarging them to the same size as the other sites might lead to unclear geographical location markings. Therefore, in the revised Figures 7, 9 and S12, we moderately increased the sizes of the wind roses for OBS and JUS.

Page 11, line 32: "from July 2020 and December 2022" should be "from July 2020 to December 2022."

**Response:** Corrected.

Figure S5: A time series of 2 years of hourly data in a figure is hard to interpret. It could be helpful to include some averaging to make it easier to read, maybe in an additional figure.

**Response:** Following the suggestion, we have revised the figure to show daily $CO_2$ concentrations instead of the hourly data. Note that Figure S5 has become Figure S7 in the revised manuscript.

---

## Author Comment (AC3)

We would like to thank the reviewer for the comments and suggestions to our manuscript. In the following, we answer to the reviewer's comments and indicate the changes in the manuscript that were implemented according to the recommendations. The comments are in black. Our answers are in blue.

**Referee #3:**

Lian et al., Development and deployment of a mid-cost $CO_2$ sensor monitoring network to support atmospheric inverse modeling for quantification of urban $CO_2$ emissions in Paris, is a well-written and thorough piece of science that will be a useful resource for others working in this burgeoning field. I have only two substantive comments that the authors might consider addressing briefly within the text.

**Response:**

We thank the referee for the positive comments on our manuscript.

Firstly, on page 5, line 17, the authors write: "measuring dry air from two target cylinders with known $CO_2$ mole fractions. ... for a duration of 10 minutes, utilizing only the last three minutes of data", when discussing the calibration. Later in that paragraph, they say, "The target gas is injected ... for a duration of 3 minutes and only the last-minute data are used" to deal with sensor drift. It would be helpful if the authors commented on the sufficiency of the three minutes of target gas measurement. Is the measurement stable after two minutes? Did the authors run the target gas for longer periods and determine this was optimal for eliminating sensor drift while minimizing gas consumption?

**Response:**

We have added the following paragraph in the supplement (Text S1 and Figure S4) to explain the settings for these two flushing times:

"To mitigate delays in sensor responses and ensure stability, thorough $CO_2$ flushing of the sensor cell is necessary. During the $CO_2$ correction coefficient $IC_1$ determination process, we sequentially sampled $CO_2$ mole fraction for a duration of 10 minutes, with 7 minutes dedicated to flushing and only the last 3 minutes of data used. During the on-site daily target gas injection for the $CO_{2_{offset}}$ calculation, we sampled $CO_2$ mole fraction for a duration of 3 minutes, with 2 minutes of flushing and only the last minute of data used.

The differences in flushing times are due to two reasons. First, the $CO_2$ correction coefficient $IC_1$ is determined through a multipoint $CO_2$ regression using the seven mole fraction values assigned within the 400-600 ppm range. Conversely, the $CO_2$ concentration in the target tank (which contains dry compressed natural air, pressurized at 200 bars and calibrated in $CO_2$) is supposed to be close to the ambient air $CO_2$ concentration on-site during midday. The step between two different $CO_2$ concentrations in the $IC_1$ determination process is greater than that during the target tank injection for drift correction, thus requiring a longer flushing time to achieve stabilization. Second, the CRDS and the mid-cost HPP sensor do not measure at the same flow rate, approximately 0.25 LPM for the CRDS and about 1 LPM for the HPP. They also have different precision targets. The CRDS sensor requires an extended period of target gas measurements to achieve a stability of less than 0.05 ppm, which is suitable for applications beyond this specific intercomparison. Therefore, the flushing time in the $IC_1$ determination process, when the HPP sensor measures in parallel with the CRDS, is expected to be longer.

Before implementing this setting, we carried out several sensitivity tests on the sensor performance with a daily injection of target gas lasting 5 minutes at LSCE laboratory. The following figure shows the evolution of target gas injection duration in relation to the differences in $CO_2$ concentration between the other 4 minutes and the 3rd minute at one HPP sensor (HPP3) over 26 days. It demonstrates that a 3-minute target gas injection, specifically utilizing the 3rd minute data, proved to be sufficient. The added value of the 4th- and 5th- minute injection is rather limited. Therefore, the choice of a two-minute flush serves as a good compromise between maintaining good sensor performance (ensuring a target accuracy of 1 ppm) and minimizing gas consumption (thereby extending the lifespan of the tank and reducing associated maintenance requirements)."

[Figure]

Figure S4. The evolution of target gas injection duration in relation to the differences in $CO_2$ concentration between the other 4 minutes and the 3rd minute at one HPP sensor (HPP3), with a daily injection of target gas lasting 5 minutes over 26 days at LSCE.

Secondly, the authors have invested a huge effort in developing, testing and deploying the HPP sensors. Much of that effort will be 'banked' e.g. the data handling investment, but there remain significant recalibration efforts when replacing the target tanks every 4-5 months. A short comment on the relative cost saving over the lifetime of the HPP sensors relative to investment in a higher-precision instrument such as a Picarro would be helpful to the audience.

**Response:**

Thank you for this valuable suggestion. We have added the following sentences in the discussion section:

"The development of mid-cost and medium-precision instruments require a certain amount of funding, manpower and time. After the 2.5-year experience in Paris, the maintenance costs for HPP instruments have been gradually decreased, and their performance has become more stable compared to the initial stages. As of now, the HPP sensor itself is performing well and operating normally. Most of the routine maintenance for the integrated HPP instrument mainly involves cleaning or replacing parts such as the micro-pump and membrane filter. We will continue to monitor the lifespan of this first generation of mid-cost instruments in order to calculate their final expenses and compare them with the high-precision CRDS instrument. In addition, we are also working on several lab developments, such as testing the dual target gas calibration strategy and assessing the impact of adding a thermo-regulated unit, in order to further improve the accuracy of mid-cost instruments. However, it should be noted that these configurations will further increase the cost of the instruments. Finding a balance between accuracy and cost,

ensuring that the number of deployed instruments meets the different needs of $CO_2$ emission monitoring for cities, and comparing these with the operational costs of high-precision CRDS instruments are all crucial considerations."

I make some minor recommendations to improve accessibility of the text.

P2, line 27: use 'the ninth' rather than 'a ninth',

**Response:** Corrected.

There is some inconsistency in the text about the use of p and P to denote pressure. Please stick to one or the other.

**Response:** Modified. We have used P consistently in the text, equation, figure and table.

p6, line 34. Change to "For a list of internal flags for some important physical parameters, refer to Table S1".

**Response:** Changed as suggested

p8. line 33-34. "The p correction substantially reduces the RMSEs of $\Delta CO_2$ to 1.6ppm (HPP4) to 49.7 ppm (HPP7)" is slightly confusing. Suggest revision to: The p correction generally substantially reduces the RMSEs of $\Delta CO_2$. For instance, in HPP4, the p correction reduces the RMSE of $\Delta CO_2$ to 1.6ppm (an improvement of 88% relative to the Raw, $H_2O$ and T corrected RMSE)." While it is commendable that the authors also cite HPP7, the relative improvement in that case was only 1%, significantly lower than the other seven sensors, which causes some confusion for the reader taking in the whole dataset.

**Response:** Revised as suggested

p22. Figure 6 is very dense. Perhaps panel b) with the modeling data could be omitted and the aspect ratio adjusted for an improved reader experience.

**Response:** As suggested, panel (b) with the modeling data was omitted, and the aspect ratio of the figure was adjusted. We also bolded the font for an improved reader experience.

p.24 Figure 8 is also very complex and hard to read. Possibly this information could be moved to the supplementary material and a smaller sub-set (possibly only winter and summer and/or every second site by distance from JUS) displayed in Fig 8, to improve readers experience.

**Response:** Following the suggestion, the original Figure 8 has been split into two separate figures. The revised Figure 8 now displays the winter and summer periods, while the newly added Figure S11 shows the spring and autumn periods.

---

## Author Response (AR2)

The authors have addressed nearly all of the reviewer comments. There is one more very minor comment from Reviewer #1 that I would like to see addressed before final publication. The comment is pasted below:

"Differences between JUS and other sites: I appreciate the added clarification from the authors, but it appears my initial comment was unclear. What I find confusing about this section is that the authors attribute small differences in $CO_2$ mole fraction between JUS and most of the other urban sites to spatial proximity, but seem to imply that a large difference between JUS and another urban site (OBS) is due to measurement error (page 12, lines 10-12 in track changes document). These two statements seem contradictory, i.e., if the measurement error means these gradients cannot be interpreted, that should apply to all of the gradients."

**Response:**

In this paragraph, we aim to show that there are small differences in $CO_2$ mole fractions at the JUS and HPP urban sites (excluding OBS), both in the observation and model data. These observed and modeled differences in $CO_2$ concentrations between urban sites are smaller than those between urban and suburban sites because of their spatial proximity, which is further linked to the spatial differences in $CO_2$ emissions.

The observed $CO_2$ difference between JUS and OBS is larger compared to their modeled difference. However, we do not imply that this large discrepancy is merely due to measurement errors. It could also be caused by factors such as inconsistencies between actual $CO_2$ emissions and those reported in inventories. Note that the colocation performance shown in Figure 5 also indicates that the RMSE for the HPP7 sensor (OBS) is comparable to that of the other HPP sensors and is not significantly larger. We have added the following sentence in the revised manuscript: "This is probably because of inconsistencies between the actual spatiotemporal distribution of intra-urban $CO_2$ emissions and those reported in the inventory."

The discussion on measurement error on page 12, lines 10-12 of the track changes document refers to all HPP sites, not just the OBS site. We have changed the word "various" to "nearly all" to better clarify it. Additionally, this point is further explained and highlighted in the conclusion section of the manuscript (See page 14, lines 6-9 in track changes document), as shown below:

"However, afternoon $CO_2$ mole fraction differences between station pairs in summer, especially the HPP stations located within the Paris city limits, are quite small, typically below 1 ppm. In these cases, the accuracy of the HPP instruments is not sufficient to identify model-observation misfits that would be generated by an error in the emission estimate in the downtown areas of Paris."